



# Recovery of sparse urban greenhouse gas emissions

Benjamin Zanger[1], Jia Chen[1], Man Sun[1], and Florian Dietrich[1]

[1]Environmental Sensing and Modeling, Technical University of Munich (TUM), Munich, Germany

**Correspondence:** Benjamin Zanger (benjamin.zanger@tum.de) and Jia Chen (jia.chen@tum.de)

**Abstract.** To localize and quantify greenhouse gas emissions from cities, gas concentrations are typically measured at a small number of sites and then linked to emission fluxes using atmospheric transport models. Solving this inverse problem is challenging because the system of equations is usually underdetermined. A common top-down approach to solving this problem is Bayesian inversion that uses a given Gaussian prior emission field. However, such an approach has drawbacks when the assumed spatial emission distribution is incorrect. In our work, we investigate sparse reconstruction (SR), an alternative reconstruction method that does not require a prior emission field, but only the assumption that the emission field is sparse. We show that this assumption is mostly true for the cities we investigated and that the use of the discrete wavelet transform helps to make the urban emission field even more sparse. To evaluate the performance of SR, we created concentration data by applying an atmospheric forward transport model to $CO_2$ emission inventories of several major European cities. We used SR to locate and quantify the emission sources by applying compressed sensing theory and compared the results to regularized least squares (LS) methods. Our results show that SR requires fewer measurements than LS methods and that SR provides better localization and quantification of unknown emitters.

## 1 Introduction

Understanding anthropogenic greenhouse gas (GHG) emissions is important for scientists and decision makers fighting climate change. Based on a growing amount of atmospheric observations, studies estimating emission fields of GHG sources and sinks from these observations have been performed on local (Chen et al. (2016); Viatte et al. (2017); Toja-Silva et al. (2017)), metropolitan (Jones et al. (2021); Turner et al. (2020); Hase et al. (2015)), country (Miller et al. (2013); Shekhar et al. (2020)), and global (Hirsch et al. (2006); Mueller et al. (2008); Turner et al. (2015); Jacob et al. (2016)) scale. One of the main reason for such studies is to verify and improve GHG emission inventories created by bottom up methods. Verification and improvements include, but are not limited to:

- Determining the difference between the real emissions and the emissions captured by inventories.

- Determining differences between the real and bottom up estimated emissions for individual emitters.

- Finding emitters which are not captured by inventories (unknown emitters).

Atmospheric inverse modeling methods use column or in situ GHG concentration measurements to estimate emission fields. Due to a lack of measurements and high modeling and measurement uncertainties, estimating each grid cell of an emission



field independently is not possible. Instead, sectors (Jones et al., 2021) or spatial correlations (Wesloh et al., 2020) are used to construct alternative parameterizations of emission fields to prevent overfitting.

An alternative to overcome these issues are sparse reconstruction (SR) methods (Ray et al., 2015). SR methods can use concentration measurements to estimate sparse emission fields, meaning that only a small number of large emitters contribute
significantly to the total emissions. These methods determine the critical emission grid cells and adjust the emissions of only those cells until the model best matches the observations. All other grid cells are set to zero. Once conditions of compressed sensing (CS) are fulfilled, SR methods are guaranteed to determine the best possible emission grids to be fitted and provides a good estimation of their emissions.

SR for the recovery of GHGs has been proposed by several recent studies. Ray et al. (2014, 2015) used stagewise orthogonal
matching pursuit (StOMP), a reconstruction method known from compressed sensing, and modified it to enforce positive emission estimates. To overcome the restriction of sparse reconstruction of only being able to reconstruct sparse fields, they used a multi scale resolution field based on wavelets. In Ray et al. (2015), StOMP has been modified so that prior information can be included. Both studies reconstructed fossil fuel $CO_2$ emissions in an idealized scenario with synthetic measurements and very low measurement noise (SNR > 40 dB). Hase et al. (2017) demonstrated sparse reconstruction with enforced positive emis-
sions estimates of anthropogenic $CH_4$ emissions from synthetic observations in the US. To overcome the limitation of sparse emission field estimations, Hase et al. (2017) used a redundant dictionary representation, where multiple parameterizations are possible for the same emission field. Yan et al. (2012) proposed compressed sensing, i.e. sparse reconstruction with guaranteed best feature selection, for environmental monitoring with the focus on undersampling.

In this paper, we apply SR for assessing urban GHG emissions. Current GHG emission monitoring systems in cities, such
as Dietrich et al. (2021); Shusterman et al. (2016) and Sargent et al. (2018), acquire GHG concentration in the atmosphere as column or in situ concentration measurements. These measurements are then related to city emissions and background concentrations, where the city emissions are the unknowns of interest while the background is (partially) known. Göckede et al. (2010) have shown that on smaller domains, uncertainties in the background has a high influence on the estimation of the city emissions. Therefore, modern approaches, such as Jones et al. (2021) and Klappenbach et al. (2021), use the
measurements acquired to additionally improve the certainty of background concentrations using a Bayesian approach. In this work, we ignore the background and make the assumption that it is known in full detail. Extending our approach to include background concentrations is straightforward.

As urban emissions, we are using anthropogenic emission inventories from multiple European cities. To overcome the sparsity constraints of SR for non-sparse emissions, we use a wavelet transformation.
We are the first to apply SR to the estimation of urban GHG emissions. The findings of our work are the following:

- Urban emissions are mostly sparse and a 3rd level wavelet transform performs well in sparsifying urban emissions further.

- SR needs less measurements than Gaussian prior methods to achieve a similar performance if the emissions are sparse enough.




– SR performs well on localizing and quantifying large emitters, leading to the application of finding unknown emitters
not captured by emission inventories.

The paper is structured as following: Sec. 2 gives a formulation of inverse problems, introduces the reader to sparse reconstruction methods, compressed sensing, and compressible emissions and provides a discription of the algorithms used in this paper. The compressability of the anthropogenic emissions in European cities is discussed in Sec. 3. Section 4 shows selected scenarios of our reconstruction method, highlighting beneficial conditions and use cases of our reconstruction method, as well as discussing measurement noise. In Sec. 5 we create case studies for different European cities in an idealized and noisy case.

## 2   Methodology

This section gives the problem statement of atmospheric inverse problems (Sec. 2.1), provides an introduction to the theory of sparse reconstruction (Sec. 2.2, Sec. 2.3, and Sec. 2.4), and introduces measures (Sec. 2.5) and algorithms (Sec. 2.6) for sparse reconstruction.

### 2.1   Inverse Problems

An inverse problem is a problem in which input parameters should be determined from the observation of a process. For the problem in this paper, those input parameters $x \in \mathbb{R}^n$ are the GHG emission fluxes for each grid cell in an emission field and the measurements $y \in \mathbb{R}^m$ are in situ or column measurements of GHG concentrations in the atmosphere. These quantities are connected by an atmospheric process, referred to as forward model $F$, $y = F(x)$. In practice, such a forward model can be a linear, non-linear, or even a stochastic process. For this analysis, we limit the forward model to linear cases. Therefore, we can write $y = Ax$, where $A \in \mathbb{R}^{n \times m}$ is called the sensing matrix. A least squares estimation of the GHG emission fluxes $x$ is given by

$$\hat{x} = \underset{x}{\arg\min} \|Ax - y\|_2^2. \tag{1}$$

Often, however, such inverse problems are ill-posed, as in the cases we deal with in this paper. In such cases, the least squares estimation does not provide a useful reconstruction technique. For a detailed discussion of ill posed problem we refer to chapter 3 of Nakamura and Potthast (2015).

### 2.2   Bayesian Inversion

A typical approach in atmospheric sciences to solve inverse problems is Bayesian inversion. In such a setup, the unknown emissions are assumed to follow a known probability distribution. This probability distribution is referred to as a priori. Measurements are used to update the a priori, which results in a new probability distribution referred to as a posteriori. From this posteriori distribution a parameter estimation can be made using a maximum likelihood (ML) detector on the a posteriori. Since the ML detector acts on the a posteriori, this is commonly referred to as Maximum a posteriori (MAP) detector. Let us call the probability distribution of the a priori $p_X(x)$. Furthermore, probability distributions are assigned both to the observations and





the model to account for uncertainties. The measurement distribution is written as $p_Y(y)$ and the distribution which maps $x$ to $y$ is written as $p_{Y|X}(y|x)$. Using Bayes' theorem, a posteriori distribution for $x$ under the condition of the observations $y$ can be derived:

$$p_{X|Y}(x|y) = \frac{p_{Y|X}(y|x)p_X(x)}{p_Y(y)}. \tag{2}$$

On this derived distribution the MAP detector can be applied. Since $p_Y(y)$ is only a constant factor for some specific measurements, the MAP detector can be written as

$$\hat{x} = \arg\max_x \ p_{Y|X}(y|x)p_X(x), \tag{3}$$

where $\hat{x}$ are the estimated physical quantities. Typically the distributions are assumed to be Gaussian, where the a priori distribution is assumed to be centered around an initial guess, $x_A$. This allows for easier error analysis and makes the problem computationally feasible.

## 2.3 From Bayesian Inversion to Regularization

To show the relation between Bayesian inversion and regularization methods, we show how a Bayesian inversion problem, using Gaussian priors, can be converted to a regularization problem. Assume that $p_{Y|X}(y|x)$ is Gaussian distributed with the covariance matrix $S_o$,

$$p_{Y|X}(y|x) = \frac{1}{\sqrt{(2\pi)^m \det(S_o)}} \exp\left(-\frac{1}{2}\left\|S_o^{-1/2}(Ax - y)\right\|_2^2\right), \tag{4}$$

where $m$ are the number of measurements, and $p_X(x)$ to be a Gaussian prior of the form

$$p_X(x) = C \exp\left(-\frac{1}{2}\left\|S_A^{-1/2}(x - x_A)\right\|_2^2\right), \tag{5}$$

where $C$ is a normalization constant. Applying the MAP detector from Eq. (3) gives an estimation of

$$\hat{x} = \arg\max_x \ p_{Y|X}(y|x)p_X(x) \tag{6}$$

$$= \arg\min_x \left[\left\|S_o^{-1/2}(Ax - y)\right\|_2^2 + \left\|S_A^{-1/2}(x - x_A)\right\|_2^2\right]. \tag{7}$$

The idea of regularization methods on the other hand is to add a penalty term to the least squares problem from Eq. (1) to prefer solutions of a certain kind,

$$\arg\min_x \left[\left\|C_1(Ax - y)\right\|_2^2 + \lambda R(C_2 x)\right], \tag{8}$$

where $R : \mathbb{R}^n \to \mathbb{R}$ is the regularization function and $C_1, C_2$ are correlation matrices. This equation is equivalent to Eq. (6) with $\lambda = 1$, $C_1 = S_o^{-1/2}$, $R = \left\|S_A^{-1/2}(x - x_A)\right\|_2^2$, and $C_2 = 1$. For such a regularization function, the regularization scheme is known as Tikhonov regularization or also ridge regression (Golub et al., 1999).





In this paper, we investigate sparse reconstruction (SR) methods. To achieve SR, the regularization term has to be changed so that sparse solutions are preferred over non-sparse solutions. In statistics, such a regularization function is the Lasso regularization function, presented by Tibshirani (1996). The Lasso is given by

$$R(x) = \sum_j |x_j| = \|x\|_1 \tag{9}$$

and is used especially when $x$ is approximately sparse. The Lasso is expected to select those elements in $x$ which are important and meaningful, while the irrelevant features are estimated to be zero. Su et al. (2017) showed that this is not necessarily the case, as coefficients which are zero in $x$ are sometimes estimated to be important (which is referred to as false discovery). In the next section, we introduce compressed sensing (CS), which provides sufficient conditions to prevent false discoveries in $x$ using Lasso regularization.

### 2.4 Compressed Sensing

Compressed sensing (CS) is a theory which provides sufficient conditions to guarantee best possible reconstruction using the Lasso regularizer, therefore, preventing false discoveries. The conditions of CS apply to the forward model $A$ and are hard to examine. For this reason, the conditions are normally already considered in the design process. In the following, we provide the very basics of CS needed to understand our work. For a more comprehensive introduction to CS, we refer to Boche et al. (2015).

CS states that an s-sparse signal $x \in \Sigma_s^n$, where s-sparse is defined as $s \geq |\{j|x_j \neq 0\}|$ and $\Sigma_s^n$ is the set of all s-sparse signals which are in $\mathbb{R}^n$, can be uniquely reconstructed by $m$ measurements $y \in \mathbb{R}^m$, defined by $y = Ax$, where $A \in \mathbb{R}^{m \times n}$, if certain conditions are satisfied for $A$. The unique solution is found by solving the $l_0$ minimization problem given by

$$\hat{x} = \arg\min_x \|x\|_0 \ \text{s.t.} \ Ax = y. \tag{P0}$$

Solving this minimization problem is NP-hard and not applicable to real world applications. Candès et al. (2006) showed that for additional conditions in $A$, one can solve the $l_1$ regularization problem instead, which is a convex problem:

$$\hat{x} = \arg\min_x \|x\|_1 \ \text{s.t.} \ Ax = y. \tag{P1}$$

Then, solving Eq. (P1) leads to the same solution as Eq. (P0). A sufficient condition for recovering (P0) with (P1) is the Restricted Isometry property (RIP) introduced in Candès and Tao (2005). This property determines a Restricted Isometry constant (RIC) $\delta_s$ for a certain sparsity level $s$, which is calculated by

$$(1 - \delta_s)\|x\|_2^2 \leq \|Ax\|_2^2 \leq (1 + \delta_s)\|x\|_2^2, \tag{10}$$

where $x \in \Sigma_s^n$. The RIC tells how close the singular values of $m \times s$ submatrices of $A$ are to 1. For $\delta_{2s} < 1$, any $s$-sparse solution can be uniquely determined by solving the (P1) problem and for $\delta_{2s} < (\sqrt{2} - 1)$ this is even possible if the signal is superimposed by noise (noisy case) (Candès, 2008). In practice, calculating the RIC is NP-hard (see Tillmann and Pfetsch





(2014)) and it is not applicable to calculate this constant for a given matrix. However, the RIP might be used within a design
process, since there are known random distributions of matrices, which satisfy the RIP for large $s$ considering sufficient large $n$
and $m$ (Baraniuk et al., 2008). Another property, which very loosely upper bounds the RIC, is the incoherence property (Wang
et al., 2015). The coherence $\mu$ of a matrix is defined by

$$\mu = \max | < a_i | a_j > | \qquad i \neq j, \tag{11}$$

where $a_i$ and $a_j$ are distinct column vectors of $\tilde{A}$, where $\tilde{A}$ is the column normalized matrix of $A$. The coherence bounds the
150 RIC by

$$\delta_{2s} \leq (2s - 1)\mu. \tag{12}$$

Therefore, $s$-sparse solutions can be uniquely recovered if $\mu < \frac{1}{2s-1}$ holds in the noiseless case or $\mu < \frac{\sqrt{2}-1}{2s-1}$ in the noisy case.

  In real world scenarios, coefficients are rarely sparse but often compressible. This means that the coefficients can be well
approximated by sparse coefficients. A more detailed explanation for compressible coefficients is found in Sec. 2.5. CS guar-
155 antees to make good estimations of compressible solutions if the RIP is satisfied for the noisy case. Then the reconstruction
error can also be bounded (see Candès (2008) for the exact definitions and bounds).

## 2.5 Sparsifying Emissions

Sparsity is one of the key elements for SR. However, emissions are not always sparse. In order to make a non sparse emission
field sparse, a transformation into a different domain can be used. Such transformations include the Fourier transform, wavelet
transform, transformations tailored to specific data sets, e.g. by SVD truncation (see Hong et al. (2011)), or overcomplete
dictionaries, where the representation does not have to be orthogonal and multiple representations for the same emission field
exist (see Candès et al. (2011)). In this paper, we only deal with the discrete wavelet transform (DWT) to sparsify emissions.
This transform is used for image compression (Lewis and Knowles, 1992) and was also used in Ray et al. (2014) and Ray et al.
(2015) to parameterize fossil fuel $CO_2$ emissions for the US.

There are several ways to quantify the sparsity of an emission field. One possibility is to measure the error, using any $l_p$
norm, of an emission field $x$ to its best s sparse approximation,

$$\sigma_s(x)_p = \inf \left\{ \| x - z \|_p, z \in \Sigma_s^n \right\}. \tag{13}$$

Independent of the norm used, the best approximation is given by an emission field $z$, which contains the same $s$ highest values
of $x$ and is zero otherwise. Often, it is more intuitive to give the relative sparsity $s_{\mathrm{rel}}$, which is the fraction of non zero entries
as a percentage of all entries, instead of the sparsity level $s$. In this paper, both notations are used, e.g. $\sigma_{10\%}(x)_2$ is the $l_2$ error
of the signal which best approximates $x$ and maximally possesses 10 % non zero elements while $\sigma_{10}(x)_2$ is the $l_2$ error of the
signal approximating $x$ with maximally 10 non zero elements. To show the distribution of values in $x$, a plot showing $\sigma_s(x)$
over all possible $s$ can be used. An example is depicted in Fig. 1. The more hyperbolic the plot is, the more compressible the
signal is. In order to get a measure for the compressability of the distribution, the Gini index is used, which was also proposed

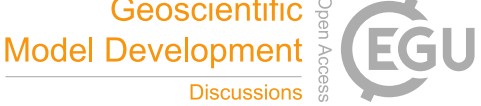

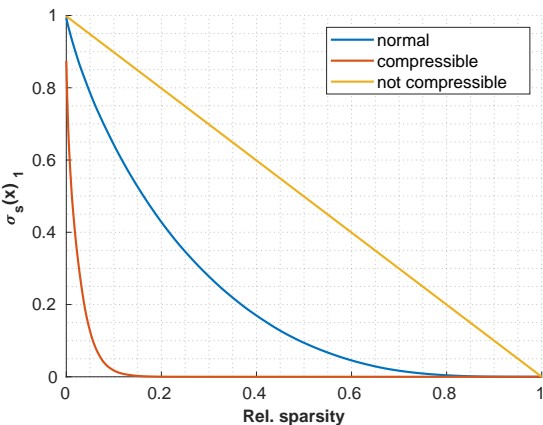

**Figure 1.** Visualization of the compressability of emission fields by plotting $\sigma_s(x)_1$ over all $s$. The more hyperbolic the curve is, the more compressible the emission field.

as a sparsity measure for CS by Zonoobi et al. (2011). This index measures how unequal $x$ is distributed, which is a similar measure as the hyperbolicity of a curve in the depicted figure. For the case of an equal distribution (yellow curve) the Gini index becomes $0$ and for a highly compressible distribution (orange curve) the Gini index gets close to $1$. A Gini index of exactly $1$ is equal to a $1$-sparse signal.

In this paper, a DWT is used to sparsify emissions, using Haar wavelets. Throughout the paper, we use the $3^{\mathrm{rd}}$ level wavelet transform and refer to its matrix as $W$ and its inverse transformation as $W^{-1}$. For an introduction to wavelets, we refer to Graps (1995). The emission map $x$ is transformed into its wavelet coefficients $c$ by $Wx = c$ and vice versa by $x = W^{-1}c$. To transform the forward model $Ax = y$ into the wavelet domain, we write

$$y = Ax = AW^{-1}c = \bar{A}c, \tag{14}$$

where $\bar{A} = AW^{-1}$ is the new forward model. The wavelet coefficients are then determined by the modified (P1) problem,

$$\hat{c} = \arg\min_c \|c\|_1 \ \text{s.t.} \ \bar{A}c = b. \tag{15}$$

Note that the CS conditions for $\bar{A}$ are not the same as for $A$. Once the wavelet coefficients $\hat{c}$ are determined, the emissions are estimated by $\hat{x} = W^{-1}\hat{c}$.

## 2.6 Reconstruction Algorithms

There have been many algorithms proposed for the task of CS and SR. However, our goal in this paper is not to compare these algorithms, instead, we want to demonstrate the applicability of SR in general for urban GHG emission assessments. We, therefore, only solve the initial SR problem, given in Eq. (P1). To find the minimum, we use the cvx library, a matlab package





for specifying and solving convex programs (Grant and Boyd (2014, 2008)), using the gurobi optimizer as a backend (Gurobi Optimization, LLC, 2021).

For the noisy case, we solve a modified form of Eq. (P1), given by

$$195 \quad \hat{x} = \underset{x}{\arg\min} \|x\|_1 \text{ s.t. } \|Ax - b\|_2^2 \leq \|\epsilon\|_2^2, \qquad \qquad \text{(P1}\epsilon)$$

where $\epsilon$ is the noise vector. This is equivalent to the Lasso with the right choice of $\lambda$. Since we generate the noise by design, the optimization process simplifies by choosing $\epsilon$ without having to find the right $\lambda$ value.

We compare our results to regularized least squares, which we refer to as least squares (LS) hereafter. The equation of the LS is given by

$$200 \quad \hat{x} = \underset{x}{\arg\min} \|x\|_2 \text{ s.t. } Ax = b, \qquad \qquad \text{(P2)}$$

in the noiseless case. There, the solution is given by the pseudoinverse. In the noisy case, the equation is given by

$$\hat{x} = \underset{x}{\arg\min} \|x\|_2 \text{ s.t. } \|Ax - b\|_2^2 \leq \|\epsilon\|_2^2. \qquad \qquad \text{(P2}\epsilon)$$

### 2.7 Data Evaluation

Table 1 gives an overview of the most important symbols and measures used in this paper. All of the measures we are using 205 for evaluating the reconstruction results compare the estimations to the true emissions, which are the city emission inventories in this paper. If the $l_p$ error is not specified, the $l_2$ error is used. The primary measure we use for the error evaluation of our estimates is the rel. $l_2$ error, since it provides a measure for the highest spatial resolution of the emissions (1 km × 1 km), while the rel. $l_2$ smoothed error shows an error for a lower spatial resolution (5 km × 5 km) and eliminates errors which are due to spatial errors. The rel. total error disregards all spatial errors and only evaluates the difference of the total estimated emission 210 to the true total emission of a city.

The mean rel. error is used if the rel. error of emission grid points independent of their contribution to the total amount of emissions is of interest. This metric is useful to evaluate how well a reconstruction method estimates emitters of a certain kind.

## 3 The Sparsity/Compressibility of Emissions in European Cities

In the following, we determine the sparsity and compressability of real world emissions. To do so, we study the $CO_2$ emissions 215 of European major cities, such as Berlin, Hamburg, Munich, London, Paris, and Vienna using data from the TNO_GHGco_v1.1 emission inventory (van der Gon et al., 2019). For the areas and species we investigate, the database provides annual grided anthropogenic emissions with a resolution of about 1 km × 1 km. Furthermore, the emissions are classified by its proxy, such as public power, industry, stationary combustion, etc. Emissions of all proxies of the dataset are included for the emission data used, making the assumption that this provides a good representation of typical emission fields for the investigated cities. We 220 measure the sparsity of the cities using the Gini index and $\sigma_{10\%}(x)_2$ in the spatial and wavelet domain. The results are given in Table 2. For all the cities we consider, the wavelet domain of the emission fields achieves a higher Gini index compared to the



**Table 1.** Table of symbols and measures used in the paper.

| Expression | Formulation | Explanation |
|---|---|---|
| **Symbols** | | |
| $n$ | $\|x\|$ | Number of unknown emissions in the model. |
| $m$ | $\|y\|$ | Number of measurements. |
| $s$ | $\|x\|_0$ | Number of non zero entries in $x$. |
| rel. sparsity / $s_{\mathrm{rel}}$ | $\dfrac{\|x\|_0}{\|x\|}$ | Number of non zero entries in $x \in \mathbb{R}^n$, relative to the size of the vector $x$. |
| $\Sigma_s^n$ | $\{x \mid \|x\|_0 \leq s\}$ | Set of all $s$ sparse vectors in $\mathbb{R}^n$. |
| **Sparsity measures** | | |
| $\sigma_s(x)_p$ | $\inf\left\{\|x - z\|_p, z \in \Sigma_s^N\right\}$ | $l_p$ error of the best possible approximation of the true emissions $x$ by a $s$ sparse emission field $z$. If $s$ is given in a percentage, $s_{\mathrm{rel}}$ instead of $s$ is used. |
| **Measures for evaluating reconstruction results** | | |
| rel. $l_p$ error | $\dfrac{\|x - \hat{x}\|_p}{\|x\|_p}$ | Relative error of how well the estimated emissions $\hat{x}$ approximate the true emissions $x$. |
| rel. smoothed $l_p$ error | $\dfrac{\|x * \square - \hat{x} * \square\|_p}{\|x * \square\|_p}$ | Relative error of smoothed reconstruction, smoothed by a periodic convolution using a 5 km $\times$ 5 km square ($\square$). |
| rel. total error | $\dfrac{\left|\sum_i x_i - \sum_i \hat{x}_i\right|}{\left|\sum_i x_i\right|}$ | Relative error of the sum of the estimated emissions $\hat{x}$ compared to the true emissions $x$. |
| mean rel. error | $\dfrac{1}{N}\sum_{i=1}^{N}\dfrac{\|x_i - \hat{x}_i\|}{\|x_i\|}$ | Mean of the rel. errors of single emission grid points. |

spatial domain. Furthermore, the approximation error of $s_{\mathrm{rel}} = 10\%$ of the signal is also lower in the wavelet domain, except for Vienna, where this error is equal for both domains. From these data, we conclude that a representation of the city emission fields in the wavelet domain should in general be better suited for SR. We also split the cities into two groups, cities which emission fields are good compressible (Berlin, Hamburg, Vienna, and Munich, Gini index > 0.7) and cities where this is not the case (London and Paris, Gini index < 0.7).

## 4 Evaluating Sparse Reconstruction of GHG Emissions

In the following, we define the estimation problem (Sec. 4.1) and apply SR to different European cities (Sec. 4.2).





**Table 2.** Sparsity of the reported emission fields in different European cities. In the wavelet domain, in all of the cases a better, or at least as good, sparse approximation of the emission map exists.

| City | number of emission fields $n$ | Gini index | Gini index DWT | $\sigma_{10\%}(x)_2$ | $\sigma_{10\%}(x)_2$ **DWT** |
|---|---|---|---|---|---|
| **not good compressible** | | | | | |
| London | 2205 | 0.544 | 0.924 | 0.359 | 0.221 |
| Paris | 2655 | 0.673 | 0.937 | 0.217 | 0.150 |
| **good compressible** | | | | | |
| Berlin | 1554 | 0.772 | 0.956 | 0.070 | 0.058 |
| Hamburg | 1554 | 0.792 | 0.956 | 0.081 | 0.066 |
| Vienna | 750 | 0.831 | 0.962 | 0.070 | 0.070 |
| Munich | 528 | 0.712 | 0.956 | 0.077 | 0.048 |

### 4.1 Formulation of the Estimation Problem

In the following, we assume that the influence of the background GHG concentration on the measurements is known in full detail. Subtracting this influence from the measurements gives the enhancement produced by the GHG emissions in the domain of interest. Therefore, the background can be ignored and the model gets simplified to a pure transport emission system. Let the sensitivity of the GHG concentration measurements $y$ to the city emissions $x$ be given by the sensing matrix matrix $A$,

$$y = Ax. \tag{16}$$

The rows in $A$ contain vectorized footprints. These footprints determine the sensitivity of the measurements to the GHG fluxes of different emission grid cells in the domain. They are determined by backward transport models, such as the Stochastic Time-Inverted Lagrangian Transport (STILT) model (Lin et al., 2003; Gerbig et al., 2003). In this paper, a simplified linear model, the Gaussian Plume model, is used as the transport model. This allows to vary parameters within the transport model without computational costly calculations of the STILT model. In Appendix B, we explain how the Gaussian plume footprints

are created and compare a footprint calculated by STILT with a Gaussian plume footprint.

Figure 2 visualizes the sensing matrix $A$, which represents the sensitivity of all measurements to each grid cell. Depending on the footprints in $A$, there might be emissions in $x$ which are not strongly sensed. The total sensitivity to a single grid cell $x_i$ is determined by the sum of the corresponding column in the sensing matrix. If this sensitivity is beneath a certain threshold, $\sum_j A_{ji} \leq \gamma$, where $\gamma = 10^{-9}$ ppm $\cdot$ km$^2$ $\cdot$ h $\cdot$ mol$^{-1}$ is the threshold, we remove the $i^{\text{th}}$ column from the sensing matrix and

245 do not reconstruct this emission grid cell. For Fig. 2 this would be the first column of the sensing matrix, where all values are 0. Physically, these removed emission grid cells are not upwind of the measurement stations and, therefore, cannot be reconstructed using the measurements.





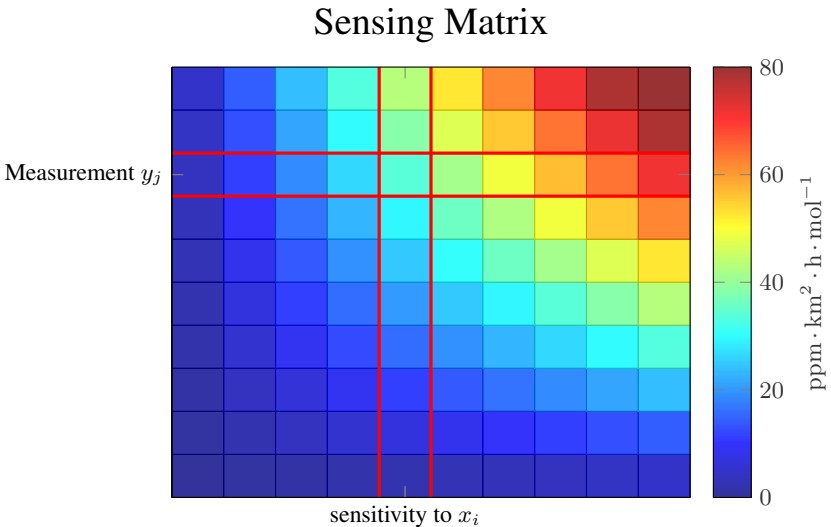

**Figure 2.** Visualization of a sensing matrix, where the values in the matrix are color encoded. The entry $A_{ji}$ in the matrix gives the sensitivity of the $j^{\text{th}}$ measurement to the $i^{\text{th}}$ grid cell.

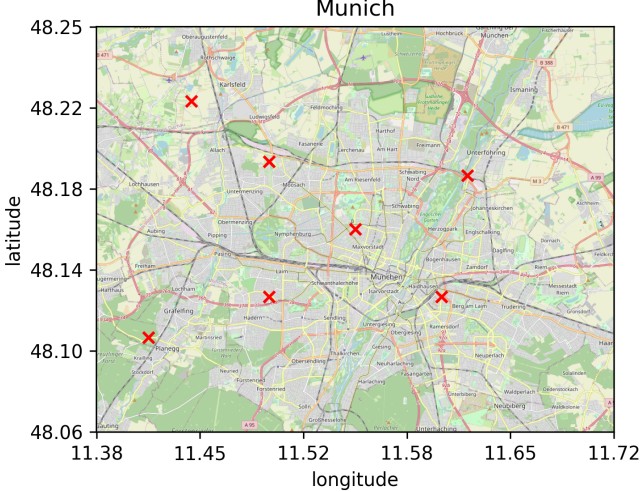

**Figure 3.** Map of exemplary measurement locations for Munich using seven measurement stations, which represents a ratio $\frac{m}{n}$ of approximately 75 % (background: ©OpenStreetMap contributors 2020. Distributed under the Open Data Commons Open Database License (ODbL) v1.0. (2020)).

In our study, we use both artificially created emission fields and real emission fields (van der Gon et al., 2019). Both fields have a spatial resolution of 1 km × 1 km. For the artificial emissions, a domain size of 32 km × 32 km is used, resulting

in $n = 1024$ unknown emission grids. For the emission inventory, the number of emission grids depends on the size of the





city (see Table 2). In order to compare emission estimates of the different domains with each other, we vary the number of measurement stations used for each domain, so that the number of measurements per unknown emission grids ($\frac{m}{n}$) is about constant. During the measurement period, each station takes 50 samples. The measurement stations were randomly distributed over a normalized area of the emission map and scaled to the domain size of each city. An exemplary distribution of the stations

for Munich is depicted in Fig. 3, using $\frac{m}{n} \approx 75\,\%$.

The measurements $y$ are then created by the sensing matrix $A$ and the emissions $x$: $y = Ax$. In the noisy case, we add a Gaussian distributed noise vector to the measurements, $y = Ax + \epsilon$, $\epsilon \sim \mathcal{N}(0, \sigma_\epsilon^2)$. $\sigma_\epsilon$ is chosen according to the SNR used.

### 4.2 Applying Sparse Reconstruction to European Cities

Reconstruction results for Munich using LS, SR, and SR in the wavelet domain are depicted in Fig. 4. From a visual inspection,

SR in the wavelet domain does resemble the real emissions best, followed by SR in the spatial domain. Negative emitters are not visible because of the color scale. LS estimates a total of 171 of such negative emitters with a total of 2139 $\mu\mathrm{mol} \cdot (\mathrm{m}^2 \cdot \mathrm{s})^{-1}$ negative emissions, while SR estimates only 33 negative emitters with a total of 9 $\mu\mathrm{mol} \cdot (\mathrm{m}^2 \cdot \mathrm{s})^{-1}$ negative emissions, and SR in the wavelet domain estimated 50 negative emitters with a total of 59 $\mu\mathrm{mol} \cdot (\mathrm{m}^2 \cdot \mathrm{s})^{-1}$ negative emissions.

In the following, we want to determine properties of SR and make comparisons to the LS using the European city emission

inventories. Those properties and differences include the interconnection between SR and CS (Sec. 4.2.1), the wavelet domain to make emissions sparser (Sec. 4.2.2), the possibility of identifying unknown emitters (Sec. 4.2.3), less measurements needed (Sec. 4.2.4), and examples including measurement noise (Sec. 4.2.5).

Additionally, in the Appendix A, we present an example using artificial, sparse emissions to better show further connections between SR and CS.

### 270  4.2.1  Influence of the Wind Coverage

We examine the effect of wind coverage on the effectiveness of the sensing matrix for SR. The term wind coverage is used to measure the range of wind directions during the measurement period. A wind coverage of $0°$ corresponds to no changes in the wind direction during the observations, while a wind coverage of $360°$ means that the wind blew from all directions during the time of observation. The wind coverage is changed in an interval from $24°$ to $360°$, with a step size of $24°$, using a Gaussian

plume model. As emission field we use the $CO_2$ city emission inventories of Munich and Paris from Sec. 3. Figure 5 depicts the rel. errors of the SR and LS reconstructions of (a) Munich's and (b) Paris's emission fields. The error of SR decreases sharply with an increasing wind coverage until a coverage of about $70°$ for Munich and $150°$ for Paris, at which the error still decreases but less sharp. In contrast, the LS has no clear improvement for higher wind coverages, but its rel. error increases with high coverages for Munich, while for Paris the error first decreases until about a coverage of $150°$ before increasing again.

This major improvement of SR for a higher wind coverage can be explained using the incoherence property from CS. The coherence parameter, given in Eq. (11), measures the maximum similarity of how two emission grid cells are measured. By increasing the wind coverage, there is a higher variety in the measurements and thus between the measurements of two different



**Figure 4.** Visualization of the (a) emission inventory of Munich and estimated emission fields of the inventory by using (b) LS, (c) SR, and (d) SR in the wavelet domain. The color scale is logarithmic and given in the unit of $\mu\mathrm{mol} \cdot (\mathrm{m}^2 \cdot \mathrm{s})^{-1}$. Negative emissions are colored dark blue and can not be visually distinguished from low or 0 emitters. The emission inventory does not include negative emitters.





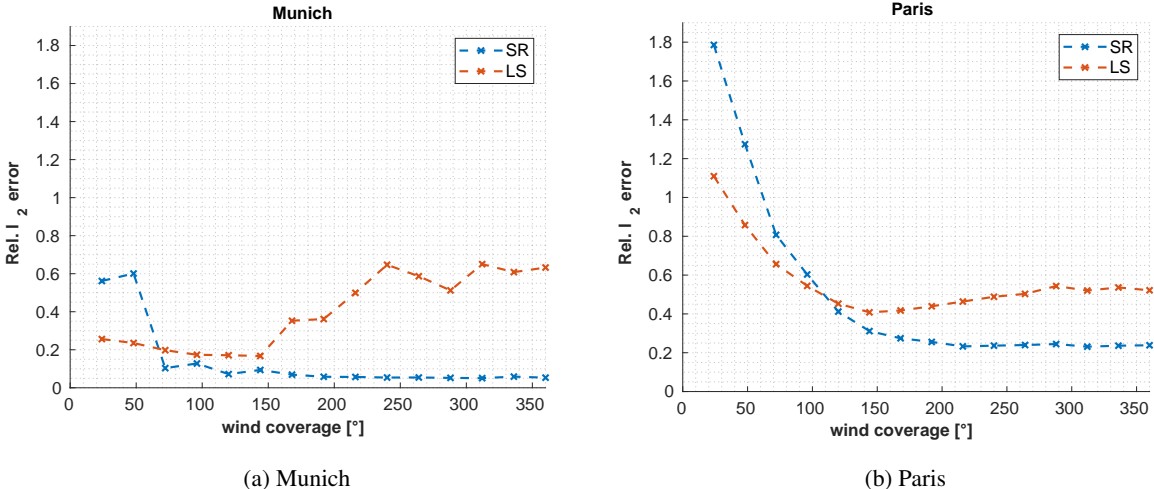

(a) Munich

(b) Paris

**Figure 5.** Reconstruction errors for SR and LS reconstruction of (a) Munich's and (b) Paris's emissions with footprints covering different amount of wind directions. With more coverages of wind directions, the reconstruction using SR improves significantly. The LS has no clear improvement for higher wind coverages, but its performance decreases with high coverages for Munich, while for Paris the performance first increases until about a coverage of 150° before decreasing again.

grid cells. Even though the coherence for a wind coverage of 360° is too high to give reconstruction guarantees, our example shows that the inclusion of CS parameters in the design process can improve reconstruction results.

### 4.2.2 Sparse Reconstruction in the Wavelet Domain

In the following, we compare reconstruction results of SR in the wavelet domain and spatial domain. For comparison of these domains, $s$ sparse city emission fields are used. More precisely, the $s$ highest emitters of the $CO_2$ inventory of London are used to generate an $s$ sparse emission field. This allows us to see at which trade-off in sparsity the wavelet domain can achieve better results compared to the spatial domain. We chose the inventory of London, since it has the least compressible emission field (see table 2).

Figure 6 shows the rel. reconstruction error in the spatial and 3$^{\mathrm{rd}}$ level wavelet domain for different sparsity levels using $\frac{m}{n} \approx 50\ \%$ and $\frac{m}{n} \approx 75\ \%$. As expected, SR in the spatial domain performs better for sparse emission maps (low $s$). For less sparse emission maps (higher $s$), however, the wavelet domain is superior.

In the case with fewer measurements, the error in the wavelet domain already plateaus for $s_{\mathrm{rel}} \approx 15\ \%$, while in the case for $\frac{m}{n} \approx 75\ \%$ the error plateaus at $s_{\mathrm{rel}} \approx 40\ \%$. The results also show that for the entire emission map of London ($s_{\mathrm{rel}} = 1$), the performance of SR in the wavelet domain does not increase significantly when the measurements are changed from $\frac{m}{n} \approx 50\ \%$ to $\frac{m}{n} \approx 75\ \%$, while the performance in the spatial domain benefits much more. This is caused by the higher compressability of the emission map in the wavelet domain, where the highest wavelet coefficients already provide a good representation of the entire emission map, while the spatial domain needs more coefficients for a representation with the same rel. error. Therefore,





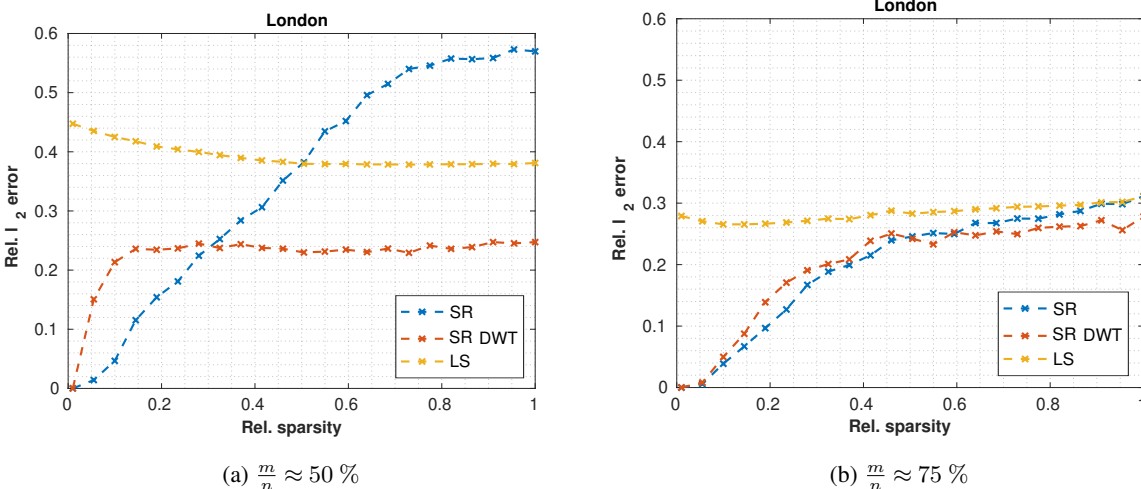

(a) $\frac{m}{n} \approx 50\,\%$            (b) $\frac{m}{n} \approx 75\,\%$

**Figure 6.** Comparison of the LS (yellow) and SR in the spatial (blue) and wavelet (red) domains in terms of rel. sparsity $s_{\mathrm{rel}}$. An emission map artificially generated from London's emission inventory with a total of 2205 unknown emitters was used. The x-axis represents the portion of these emitters that were used (the emitters are sorted from large to small). In (a) 22 measurement stations with a total of 1100 measurements and (b) 33 measurement stations with a total of 1650 measurements are used.

wavelet domain needs less measurements compared to the spatial domain to reach the same performance for non compressible cities (similar results see section 4.3.3).

Our result demonstrates that the wavelet domain can help to improve SR in those cases for which the spatial domain is not well suited. However, changing the domain also changes the conditions of CS. Therefore, conclusions made for the spatial domain regarding CS can not be directly transferred to the wavelet domain. For example, while a higher wind coverage is

beneficial for the spatial domain, this might not be the case for the wavelet domain.

### 4.2.3 Discovering Unknown Emitters

In the following, we want to evaluate SR for the task of determining unknown emitters. Assuming a good prior guess of the emission field, where the rel. error to the real emissions for each emitter is about constant and small, unknown emitters can be assumed to have a huge contribution in the difference between the prior expected measurements and real measurements.

Therefore, we evaluate how well the highest emitters are reconstructed to assess the performance of finding unknown emitters.

For this purpose, we consider the emission inventory data for Munich and Paris. While Munich's emissions inventory is good compressible, the emission inventory of Paris is not. For the setup, seven measurement stations in Munich and 39 in Paris are used. This results in $\frac{m}{n} \approx 75\,\%$ for both cities, making the estimation results comparable.

We make use of a qualitative and a quantitative measure that are both depicted in Fig. 7. The qualitative measure (see upper

panels in Fig. 7) compares the fraction of the highest emitters in the inventory agreeing with those of the estimation (the higher ratio, the better), using a logarithmic scale. We are using such a scale, because the emission field is good compressible. This





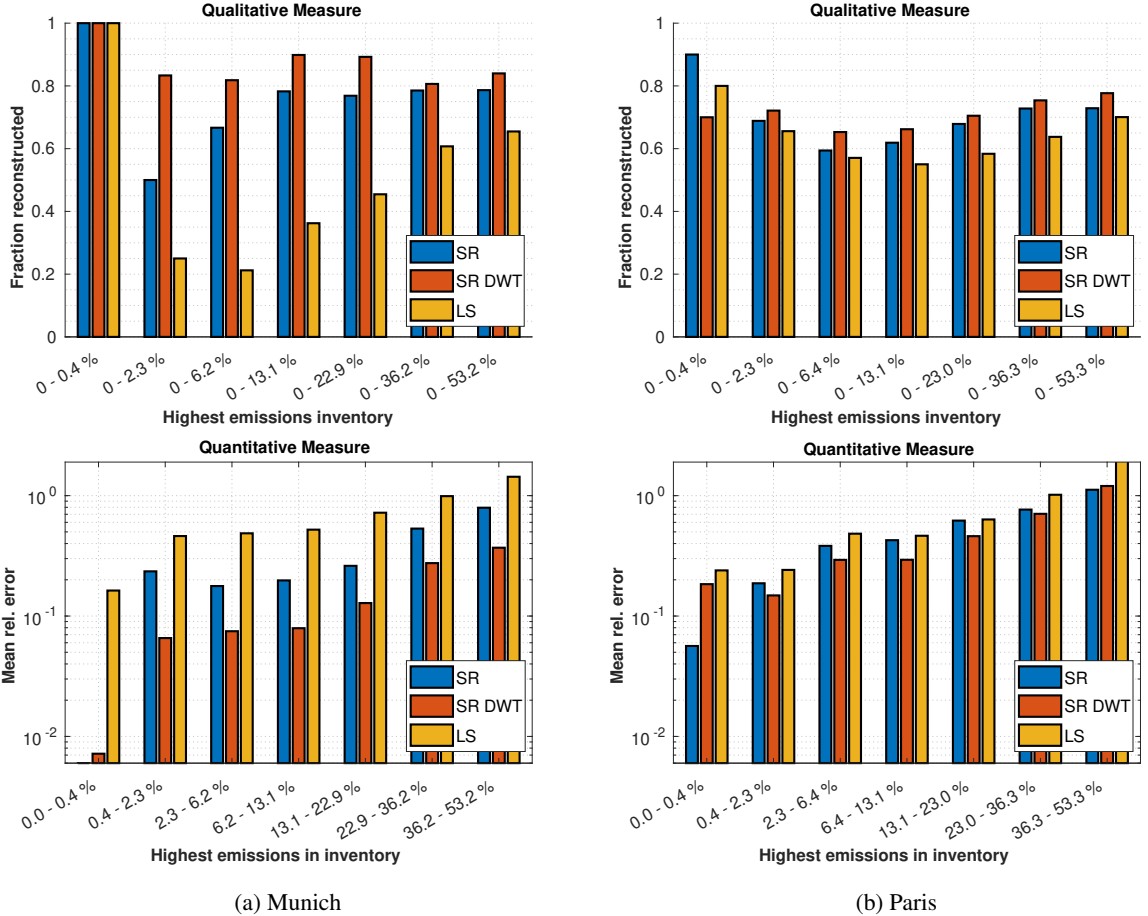

(a) Munich                                                    (b) Paris

**Figure 7.** Qualitative and quantitative measure of how well SR and LS estimate the highest emissions for (a) Munich and (b) Paris. The qualitative plot shows how many percents of the highest emissions in the inventory are also contained in the same highest amount of emission of the reconstruction. The quantitative plot shows the mean rel. errors for different emission strengths.

means that the first few emitters already have a significant contribution to the total emissions. In Munich, all three methods reconstruct the same largest $0.4\%$ emitters. For Paris, SR in the spatial domain performs best followed by the LS in case of the largest $0.4\%$ emitters. For lower emitters, SR in the wavelet domain performs best for both cities. While the performance

gain of SR and SR DWT over the LS method is very large in Munich, which has a compressible emission inventory, the gain is smaller in Paris, which has a non compressible inventory.

The quantitative measure (see lower panels in Fig. 7) compares the rel. error in the reconstructed emissions (lower the better).

For the largest $0.4\%$ emitters, SR in the spatial domain performs best and can reconstruct these emissions with an error smaller than $0.7\%$ in Munich and $3\%$ in Paris. Considering slightly lower emissions, SR in the spatial domain performance

drops significantly and the wavelet domain does the best job in both cities.



The results indicate that SR in the spatial domain works particularly well for the highest emitters (largest $0.4$ % in our examples), while SR in the wavelet domain performs well for a broader range of high emitters. The LS performance is much less sensitive to the emission strength of single strong emitters, which is not beneficial for estimating large unknown emitters.

These results for the same scenario but with less measurement stations ($\frac{m}{n} \approx 50$ %) are found in Appendix D.

### 4.2.4 Number of Measurements needed

Previous scenarios have shown comparisons between SR and LS using $\frac{m}{n} \approx 50$ % or $\frac{m}{n} \approx 75$ %. Unanswered, however, is the question how much measurements $m$ are needed to achieve descent results. To determine this, we vary the number of measurements per station, which leads to a variation of $\frac{m}{n}$.

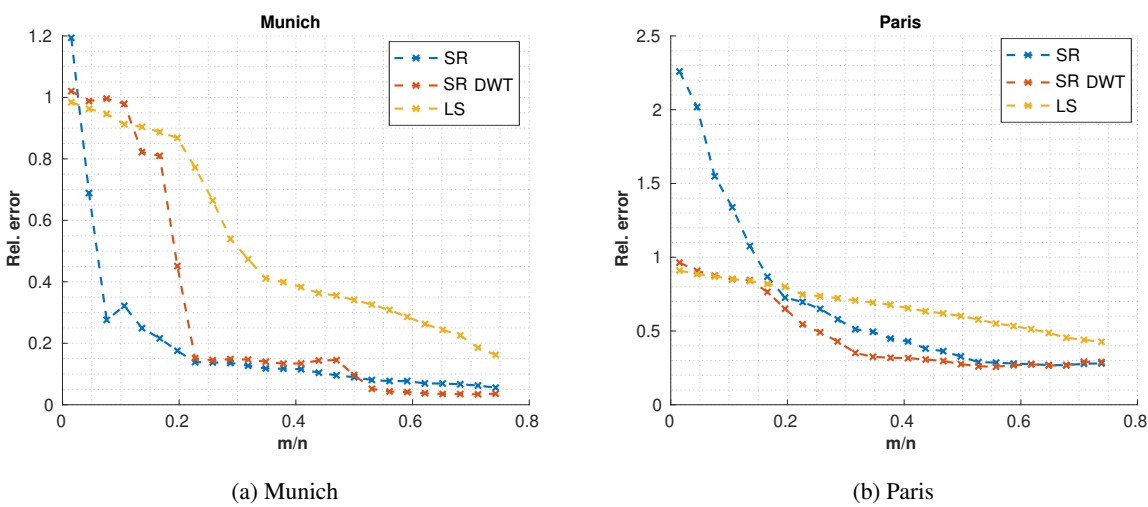

(a) Munich                                      (b) Paris

**Figure 8.** Rel. error for SR, SR with DWT, and LS at different degrees of undersampling, for (a) Munich and (b) Paris.

Figure 8 depicts the rel. estimation error for Munich and Paris for varying $\frac{m}{n}$. In Munich, where the emissions are more compressible compared to Paris, the reconstruction error for SR in the spatial domain already drops sharply for using $\frac{m}{n} \gtrsim 10$ %. SR in the wavelet domain needs more measurements but achieves similar performance as in the spatial domain using $\frac{m}{n} \gtrsim 23$ % and performs even slightly better than the spatial domain for $\frac{m}{n} > 50$ %. The LS also sees an steep improvement with $\frac{m}{n} \approx 30$ %, but does not achieve as good results as SR.

For Paris, where the emissions are not well compressible, SR in the spatial domain performs worst for $\frac{m}{n} < 20$ % while the wavelet domain performs about the same as the LS for these amount of measurements. For a higher number of measurements ($\frac{m}{n} > 0.2$), both SR methods show a performance increase and both perform better than the LS. Overall, SR in the wavelet domain performs best.

These trends are also confirmed for the other city emissions (see supplement). These results show that using SR instead of the LS, a higher undersampling with fewer errors is possible, especially for compressible emissions (as in Munich).





### 4.2.5  Measurement Noise

For making the results applicable to real world scenarios noise has to be considered, too. There are different types of noise, such as measurement noise, transport error, and representation error. In the following, we consider measurement errors with an SNR typical for column measurements. Chen et al. (2016) states measurement errors for column measurements for an averaging time of 10 minutes for $CO_2$ as $0.04$ to $0.05$ ppm and for $CH_4$ as $0.2$ ppb. Jones et al. (2021) shows the $CH_4$ concentration

enhancements measured for the city of Indianapolis are in the order of several ppb. In the following, we assume an SNR of $20$ dB which corresponds to enhancements of $2$ ppb for methane or $0.4$ to $0.5$ ppm for $CO_2$. As emission fields, we use the inventory data for Munich. Seven measurement stations with a total of 350 measurements are used, resulting in $\frac{m}{n} \approx 75\%$. We provide a variance map for SR with an SNR of $20$ dB in Appendix E. Here, we show the rel. error of the estimated emission

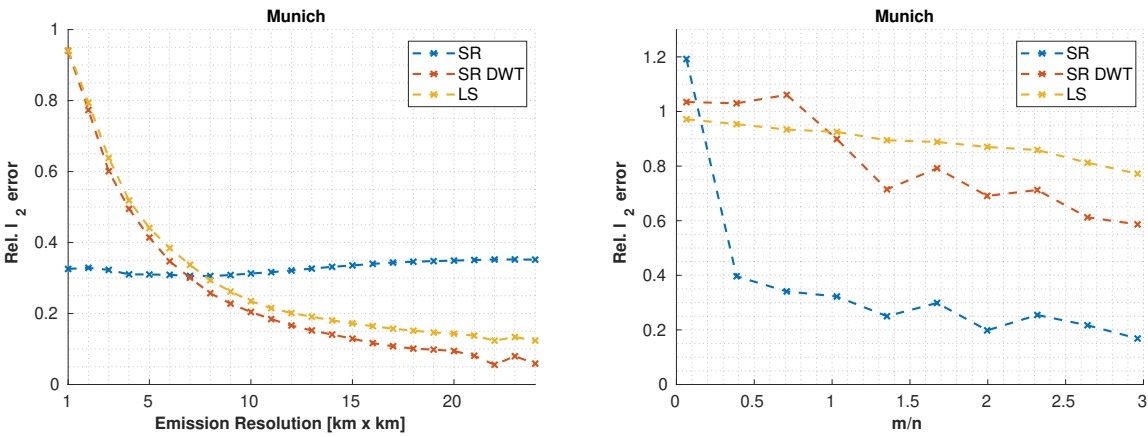

**Figure 9.** Reconstruction errors uses SR in the spatial and wavelet domain as well as the LS for Munich with an SNR of $20$ dB where (a) shows the rel. error at different levels of resolutions using $m/n \approx 75\%$ and (b) the rel. error for the highest resolution (1 km × 1 km) for varying the amount of measurements.

fields for the noisy case at different spatial resolutions of the emissions, both for SR in the spatial domain and wavelet domain

and for the LS (see Fig. 9, left). Furthermore, we assess the undersampling capability in the noisy case (see Fig. 9, right). For SR in the spatial domain, the rel. error stays fairly constant across all spatial resolutions, suggesting that almost all errors occur in the reconstruction of emission strength rather than in the localization of emitters. Both SR in the wavelet domain and the LS have a high rel. error for high resolutions (below 8 km × 8 km), while on lower resolutions both perform better than SR in the spatial domain. This indicates that most of the error is due to wrong localization of emitters.

While SR in the spatial domain performs well in localizing emitters, both for SR in the wavelet domain as well as for LS overfitting occurs and they do not provide accurate localization of the emitters. To overcome this, SR in the wavelet domain and LS need more measurements. To determine the amount of measurements needed, we show the rel. errors for the highest resolution (1 km × 1 km) for varying the amount of measurements in Fig. 9 (b). For SR in the spatial domain, the rel. error drops significantly for $\frac{m}{n} < 50\%$ and performs better compared to the other reconstruction methods for more measurements.



The wavelet domain performs similar to the LS for $\frac{m}{n} < 1$ and performs better for $\frac{m}{n} > 1$. The results indicate that SR in the spatial domain, for Munich, is more robust against noise, while the wavelet domain is not as robust (many more measurements needed than in the noiseless case for similar results).

## 5   Broader Comparison of Sparse Reconstruction for European Cities

In the following, case studies on emission fields of all cities considered in Sec. 3 are given to provide a broader comparison of the performance of SR. The measurement configuration corresponds to the specifications in Sec. 4.1. Reconstruction of emissions using SR in the spatial and wavelet domain as well as the LS is performed; first with $\frac{m}{n} \approx 75 \%$ in the noiseless case and then with $\frac{m}{n} \approx 1.5$ in the noisy case. To compare the performance, we measure rel. errors, rel. smoothed errors (for

**Table 3.** Reconstruction performances for SR in the spatial and wavelet domain as well as for LS by measuring the rel. error (1 km × 1 km), rel. smoothed error (5 km × 5 km), and the rel. total error for different European cities. The best approach for each city and type of error is highlighted in gray.

| Case study | rel. total error | | | rel. smoothed error (5 km × 5 km) | | | rel. error (1 km × 1 km) | | |
|---|---|---|---|---|---|---|---|---|---|
| | SR | SR DWT | LS | SR | SR DWT | LS | SR | SR DWT | LS |
| **not good compressible** | | | | | | | | | |
| London | 0.0 % | 0.9 % | 0.0 % | 4.5 % | 4.3 % | 4.2 % | 32.2 % | 27.8 % | 31.5 % |
| Paris | 0.0 % | 1.4 % | 0.0 % | 4.4 % | 6.4 % | 6.7 % | 27.4 % | 29.4 % | 43.2 % |
| **good compressible** | | | | | | | | | |
| Berlin | 0.0 % | 0.7 % | 0.2 % | 2.2 % | 2.7 % | 18.5 % | 8.1 % | 8.1 % | 59.0 % |
| Hamburg | 0.0 % | 1.0 % | 0.4 % | 1.9 % | 2.3 % | 15.0 % | 8.2 % | 8.8 % | 62.2 % |
| Munich | 3.1 % | 1.3 % | 4.3 % | 3.5 % | 2.4 % | 10.9 % | 6.1 % | 3.9 % | 26.2 % |
| Vienna | 2.2 % | 1.5 % | 1.3 % | 4.9 % | 3.2 % | 13.0 % | 6.9 % | 6.2 % | 48.8 % |

a spatial resolution of 5 km × 5 km), and rel. total errors. The results for the noiseless case are given in Table 3. At the highest spatial resolution (1 km × 1 km, right column in Table 3), SR performs much better than LS for the well compressible emission fields in both domains. For the cities with emission fields that are not well compressible, the SR performance is significantly worse compared to the other cities but still slightly better than LS. For a lower spatial resolution of 5 km × 5 km (middle column in Table 3), the error of all methods decreases. Among the good compressible emissions, SR still performs significantly better than LS. Both for London and Paris (not good compressible), all methods produce a very similar error, with the LS giving the smallest error in London and SR in the spatial domain giving the smallest error in Paris. For the total emissions, all reconstruction methods perform very similarly.

These results support our previous findings that SR performs well at high resolutions for cities with good compressible emissions.





**Table 4.** Reconstruction performances for SR in the spatial and wavelet domain as well as for LS by using $\frac{m}{n} \approx 1.5$ with an SNR of 20.0 dB for different European cities. The results are evaluated using the rel. error (1 km × 1 km), rel. smoothed error (5 km × 5 km), and the rel. total error. The best approach for each city and type of error is highlighted in gray.

| Case Study | rel. total error | | | rel. smoothed error (5 km × 5 km) | | | rel. error (1 km × 1 km) | | |
|---|---|---|---|---|---|---|---|---|---|
| | SR | SR DWT | LS | SR | SR DWT | LS | SR | SR DWT | LS |
| **not good compressible** | | | | | | | | | |
| London | 12.5 % | 1.6 % | 2.2 % | 38.4 % | 19.9 % | 19.9 % | 162.3 % | 81.7 % | 81.6 % |
| Paris | 11.8 % | 2.2 % | 1.9 % | 32.5 % | 24.0 % | 24.4 % | 144.4 % | 83.9 % | 83.1 % |
| **good compressible** | | | | | | | | | |
| Berlin | 10.6 % | 0.1 % | 1.3 % | 18.0 % | 29.0 % | 32.6 % | 36.5 % | 85.3 % | 85.8 % |
| Hamburg | 10.1 % | 1.2 % | 1.5 % | 20.7 % | 32.7 % | 34.1 % | 70.3 % | 88.2 % | 87.0 % |
| Munich | 19.3 % | 2.6 % | 5.6 % | 16.9 % | 23.6 % | 32.8 % | 21.4 % | 70.8 % | 84.2 % |
| Vienna | 11.2 % | 21.6 % | 6.3 % | 20.2 % | 61.5 % | 41.4 % | 38.9 % | 96.7 % | 92.3 % |

Next, we consider the noisy case with an SNR of 20.0 dB and $\frac{m}{n} \approx 1.5$. The results are depicted in Table 4. For the cities that are not well compressible, SR in the spatial domain performs the worst, while SR in the wavelet domain and the LS perform very similar.

For the well compressible cities, SR in the spatial domain achieves the best results for the 1 km × 1 km and 5 km × 5 km resolution and performs worse compared to the other reconstruction methods for the total emissions (except for Vienna). Furthermore, SR in the wavelet domain performs better than the LS. In contrast to the noiseless case, SR in the wavelet domain performs worse compared to SR in the spatial domain for high resolutions ($\lesssim$ 5 km × 5 km). Both results support our findings from Sec. 4.2.5, that SR in the spatial domain achieves a good localization of emissions and that SR in the wavelet domain is not as robust to noise as the spatial domain for high resolutions.

## 6 Conclusions

In this paper, we introduced sparse reconstruction (SR) as a novel method for the inversion of urban GHG emissions, and further provided key examples for understanding the advantages of SR to inversely model emissions.

SR can be easily integrated into existing top-down frameworks for emission estimates. We examined the applicability of this method by evaluating the sparseness of emissions from different European cities. Our results indicate that the emissions from most of the investigated cities are sparse and that SR is well applicable. We also showed that a wavelet transform increases the sparsity of urban emissions, making SR applicable to cities with less sparse emissions. SR is known to have reconstruction guarantees if conditions of compressed sensing (CS) are satisfied. We tested different CS conditions for various wind fields and showed that wind fields with better CS conditions do increase the performance of SR significantly.



Compared to state-of-the-art inversion methods using Gaussian priors, our method requires less measurements and provides better localization and quantification of unknown emitters.

SR works best if the underlying representation of the emissions is sparse. In this paper, we employed the wavelet transform to increase the sparsity. However, other transformations, such as a curvelet transform or more general dictionary representations, could be even more suited for specific spatial domains. Finding such transformations which also work well with CS conditions is challenging, and it would be interesting to study them in future.

*Code and data availability.* The code for this paper (Zanger et al., 2022b) is written in Matlab 2021a and available on GitHub: https:// github.com/tum-esm/recovery_of_sparse_urban_greeenhouse_gas_emissions (last access: 25 January 2022). The Gaussian plume footprints are available at https://doi.org/10.5281/zenodo.5901298 (Zanger et al., 2022a). The TNO_GHGco_v1.1 emission inventory (van der Gon et al., 2019) is not publicly available. The latitude and longitude boundaries used to generate the city emission fields from the inventory is given in Appendix A.

*Acknowledgements.* We would like to thank Alihan Kaplan from the chair of Theoretical Information Technology for his comments and useful discussions. BZ, JC, FD are supported by the Deutsche Forschungsgemeinschaft (DFG, German Research Foundation) (grant nos. CH 1792/2-1, INST 95/1544).

## Appendix A: Boundaries of the emission inventories

**Table A1.** Longitude and latitude boundaries used to create the $CO_2$ emission inventories for the European cities from the TNO_GHGco_v1.1 emission inventory.

| City | Southernmost latitude | Northernmost latitude | Westernmost longitude | Easternmost longitude |
|---|---|---|---|---|
| **not good compressible** | | | | |
| London | 51.30° | 51.70° | -0.52° | 0.21° |
| Paris | 48.62° | 49.10° | 1.94° | 2.66° |
| **good compressible** | | | | |
| Berlin | 52.34° | 52.68° | 13.07° | 13.66° |
| Hamburg | 53.37° | 53.71° | 9.67° | 10.27° |
| Vienna | 48.07° | 48.31° | 16.12° | 16.52° |
| Munich | 48.06° | 48.25° | 11.36° | 11.72° |

Table A1 shows the longitude and latitude boundaries used to create the $CO_2$ emission fields for the European cities from the TNO_GHGco_v1.1 emission inventory.





## Appendix B: Generation of the Sensing Matrix

The sensing matrices we use consist of vectorized footprints generated by the Gaussian plume model with artificial wind and
420 diffusion data. The Gaussian plume model is not a sufficient model for real inversion setups, but provides a good enough approximation for the purpose of this paper. The footprints are generated for a high resolution grid using a dynamic Gaussian plume with varying wind speed, direction and diffusion. The footprints are then scaled to the domain size of the city. Because of the scaling, the wind speed and diffusion changes for each city. Nevertheless, we think that this approach makes the reconstruction performances for different cities more comparable than using different footprints for every city.

For Sec. 4.2.1, we use a static Gaussian plume model to generate footprints with a different wind coverage. This allows us to ignore additional effects from the dynamical case, for example, in the dynamic case the change in the wind direction produces spiral looking footprints. These footprints have a higher sensitivity to the emission grid compared to a footprint with a lower wind coverage. By using a static model, the footprints all have a similar sensitivity to the emission grids, which makes the results more comparable to each other.

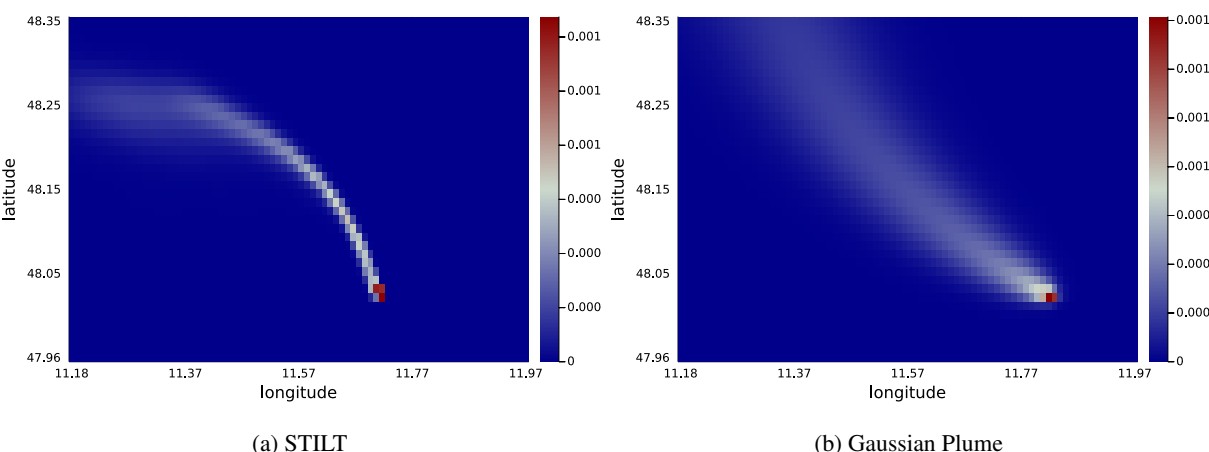

| (a) STILT | (b) Gaussian Plume |

**Figure B1.** Comparison of a STILT footprint to a Gaussian plume footprint, where different wind fields have been used.

Fig. B1 shows a comparison between a STILT footprint for Munich and a Gaussian Plume footprint used in our work, where the two footprints are generated using different wind fields.

## Appendix C: Recovery of Sparse Emissions

Consider $s$ emitters producing an $s$ sparse emission field in a city. We distribute these emitters over the city using a Gaussian distribution, such that the probability for an emitter in the center of the city is higher compared to outside of the city. Further-
435 more, we use a Gaussian distribution to model the emission strength of each emitter with a mean value of $10 \, \mu\text{mol} \cdot (\text{m}^2 \cdot \text{s})^{-1}$ and a standard deviation of $1 \, \mu\text{mol} \cdot (\text{m}^2 \cdot \text{s})^{-1}$, so that all emitters have roughly a similar contribution to the total emissions. In

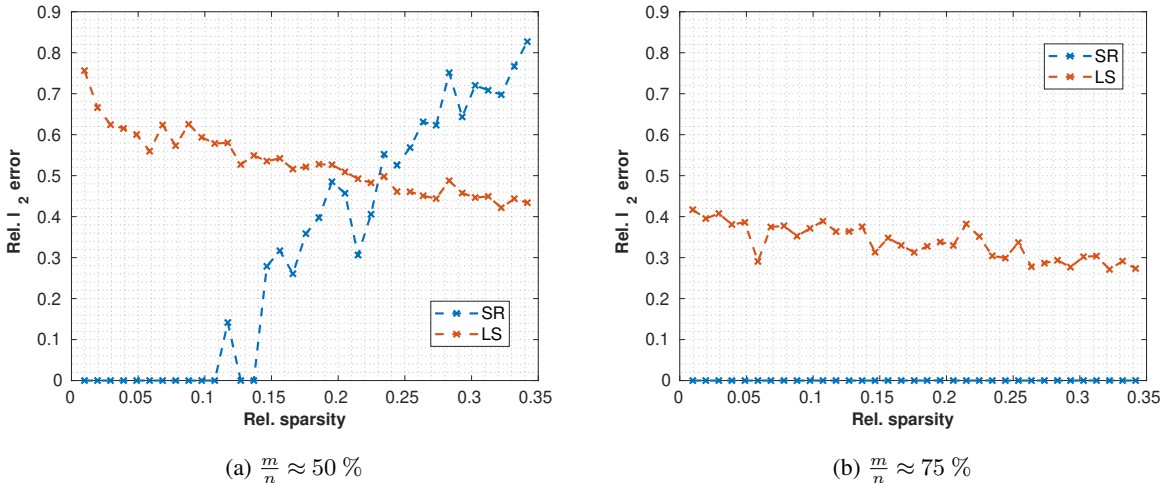

(a) $\frac{m}{n} \approx 50\,\%$                 (b) $\frac{m}{n} \approx 75\,\%$

**Figure C1.** Reconstruction errors for sparse cities with (a) 50 % and (b) 75 % measurements $m$ per emission fields $n$. Using SR (blue line), it is possible to reconstruct sparse emissions. By increasing the number of measurement stations, and therefore observations, more emitters can be reconstructed using SR (left: up to 13 %; right: up to 35 % non-zero emission cells). For the LS method (red line), the rel. error decreases both for less sparse signals and with a larger number of observations.

Fig. C1, the reconstruction errors for the spatial domain using SR and LS are depicted. SR allows exact recovery for $s_{\mathrm{rel}} \lesssim 13\,\%$ non zero emissions using $\frac{m}{n} \approx 50\,\%$ and $s_{\mathrm{rel}} > 35\,\%$ non zero emissions using $\frac{m}{n} \approx 75\,\%$. In contrast, the rel. error of LS is generally much higher, but decreases with a larger number of measurements.

These results demonstrate the power of SR for sparse emission fields and illustrate the link between SR and CS. As shown in Sec. 2.4, the sensing matrix $A$ must satisfy some sufficient conditions to ensure that CS yields the best possible reconstruction. However, in real-world scenarios, it is challenging to verify these conditions. By using a Monte-Carlo algorithm, we found counterexamples that showed $\delta_{2s} > 1$ for already small $s$ (both for Gaussian Plume footprints and STILT footprints). Thus, the sensing matrix we use here does not guarantee CS for all cases. Nevertheless, $A$ can be useful if it satisfies the CS conditions

for some specific emission distributions of interest.

    Next, we consider the case with $\frac{m}{n} \approx 75\,\%$ and chose a number of emitters $s$, such that the rel. error produced by SR and the LS is similar. Therefore, both methods produce the same rel. error, which allows us to compare the spatial distribution of the error on an equal basis. This is the case for $s_{\mathrm{rel}} = 46.9\,\%$, where the rel. error for SR is $28.1\,\%$ while the rel. error for LS is $27.3\,\%$. We focus on the following questions:

– which emitters are found correctly (discovered)?

    – which emitters are not found (undiscovered)?

    – which emitters are discovered, even though there is no emitter there (false discovered)?





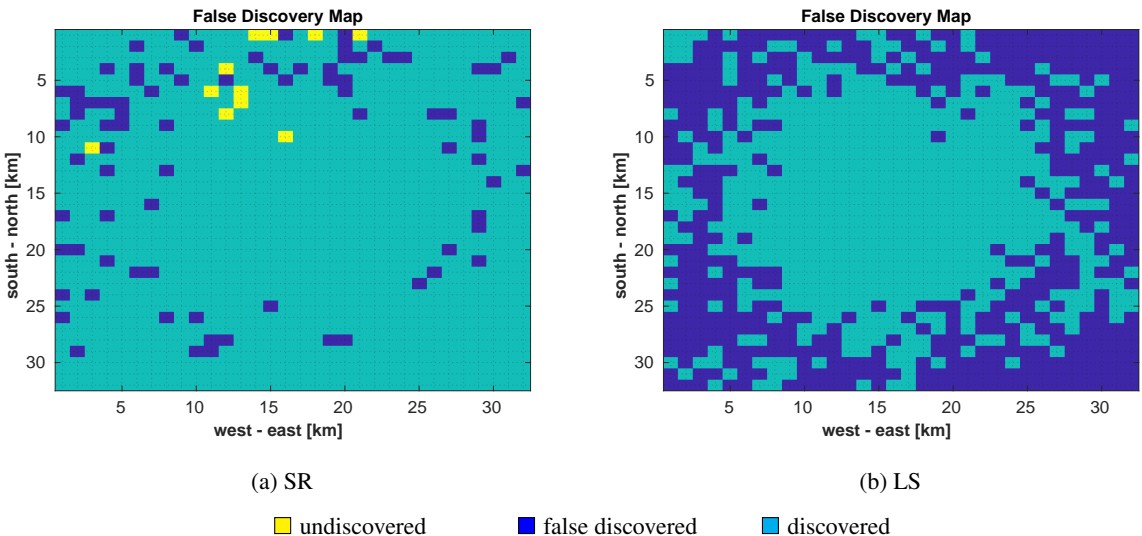

**Figure C2.** Visualization of false discovered (dark blue), undiscovered (yellow), and discovered (cyan) emissions in $x$ by (a) SR and (b) LS.

Figure C2 depicts the discovered, not discovered, and falsely discovered emitters for SR (left) and LS (right). SR reconstructs most emitters correctly, only $1.1\,\%$ of the emitters are undiscovered (yellow) and $8.0\,\%$ are falsely discovered (dark blue). In
contrast, the LS has no emitters which have not been found, but $45.6\,\%$ falsely discovered emitters. This is because the LS prefers smooth estimates, which is equivalent to estimating emitters everywhere. This is also seen by the fraction between the emissions estimated for falsely discovered emitters and the total estimated emissions. While this fractions is only $3.0\,\%$ for SR, for LS this fraction is $12.2\,\%$.

These results demonstrates the advantage of SR for the localization of emitters. This property is especially useful to find
unknown emitters, since the spatial certainty of emitters for SR is much higher.

**Appendix D:  Discovering unknown emitters using less measurements**

In Sec. 4.2.3 we showed that SR is better in finding high emitters compared to LS and can reconstruct those with smaller errors. This is visualized by the Fig. 7 which has been creates using $\frac{m}{n} \approx 75\,\%$. In the following, we resemble this result using only $\frac{m}{n} \approx 50\,\%$.





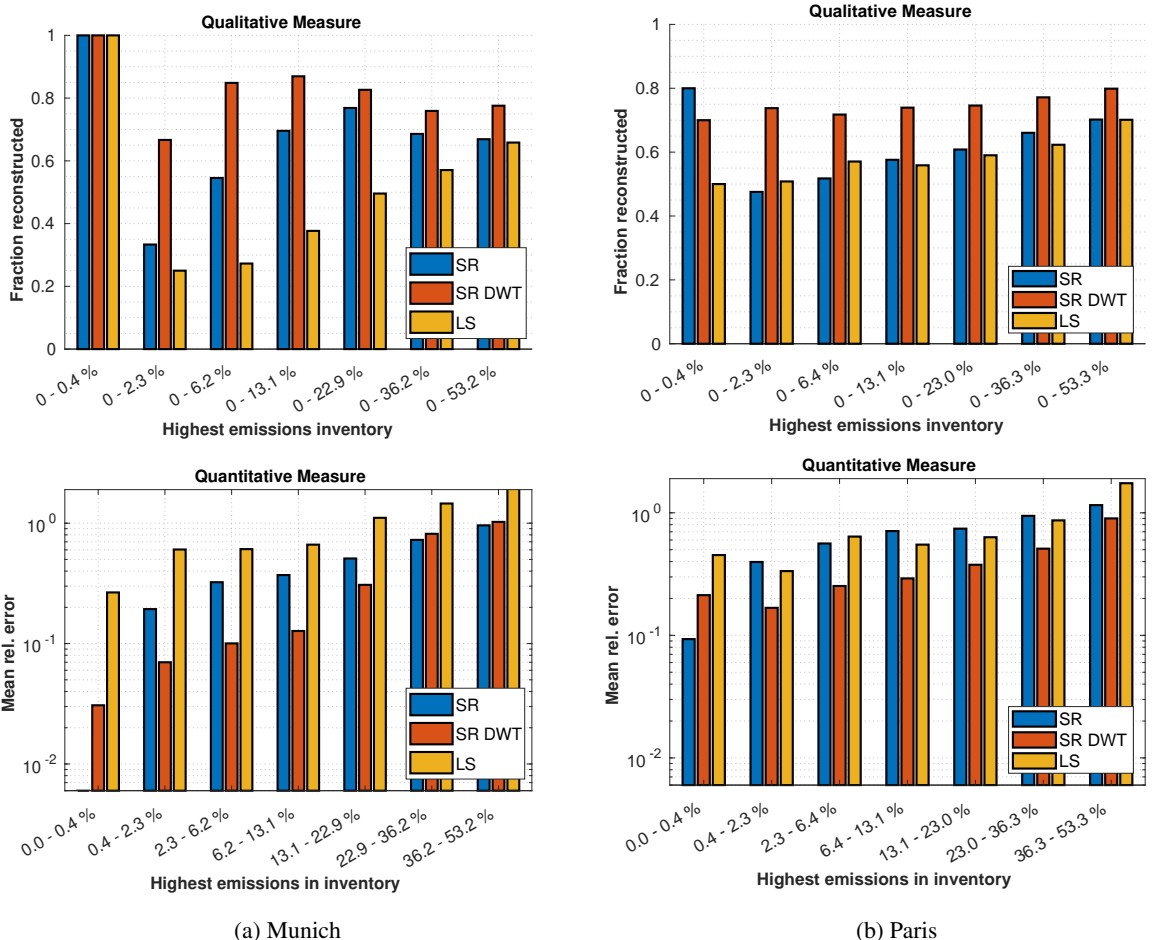

**Figure D1.** Qualitative and quantitative measure of how well SR and LS reconstruct the highest emitters for Munich (a) and Paris (b) using $\frac{m}{n} \approx 50~\%$. The qualitative plot shows how many percent of the highest emissions in the inventory are also contained in the same highest amount of emission of the reconstruction. The quantitative plot shows the mean rel. errors for different emission strengths.

The results are depicted in Fig. D1. Compared to the case with more measurements, for Munich, SR in the spatial domain performs better in the qualitative measure compared to the wavelet domain. Furthermore, for Paris, the margin between the performance of SR in the wavelet domain and the other methods has increased.

## Appendix E: Uncertainty Quantification

Uncertainty quantification for sparse reconstruction is distinct different compared to uncertainty quantification for LS recon-
struction. First, while for the LS fit with Gaussian noise a closed form solution exists, there doesn't exist any closed form solution for sparse reconstruction.



In the following we use the uncertainty quantification approach from Hase et al. (2017) in order to derive a variance plot of the reconstruction result by applying bootstrapping. This gives us the covariance matrices for SR in the spatial and wavelet domain. We only use the variances of the matrix and plot the standard deviations for every single reconstructed emission, which is depicted in Fig. E1. For the spatial domain, the standard deviation is localized around certain areas in the map, while for the wavelet domain, the standard deviation covers much larger areas of the map. This let one believe that a reconstructed emission in the spatial domain has a higher certainty of location compared to SR in the wavelet domain. This assumption is verified by Fig. 9

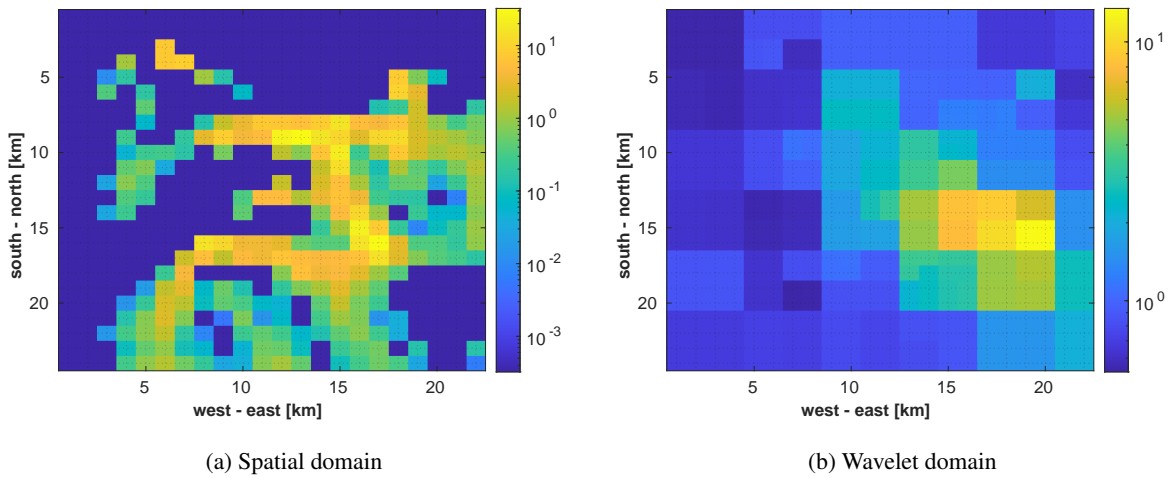

(a) Spatial domain             (b) Wavelet domain

**Figure E1.** Uncertainty of the emission estimates for Munich with an SNR of 20 dB for SR in the (a) spatial domain and (b) wavelet domain.

*Author contributions.* JC and FD conceived the study; BZ and MS performed the data analysis supervised by JC and FD; BZ, JC and FD wrote the manuscript; MS contributed to the content of the manuscript.

*Competing interests.* The authors declare that they have no conflict of interest.



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
