# Peer review of "Recovery of sparse urban greenhouse gas emissions"

_Geoscientific Model Development, 2021_

## Referee Comment (RC2)

**Referee comment**

anonymous author

**1 General comments**

The study *Recovery of sparse urban greenhouse gas emissions* addresses the problem of localizing and quantifying greenhouse gas emissions in urban settings. The authors compare different approaches to solve the inverse problem that arises when an unknown emission field is estimated by local (column) observations of trace gas enhancements. The reconstruction quality of the methods is analyzed in several synthetic and idealized settings. The theoretical study can be considered as a preparational work for current projects to monitor trace gas emissions from cities.

The content of the study is presented in a structured form and the clear language is easy to follow. In some places the description of the experimental setup is incomplete.

The study uses a simplified atmospheric transport model in an idealized setting to create a test environment for inverse methods. Modern inverse modeling approaches, sparse reconstruction and sparse reconstruction in the wavelet domain, are compared to the standard method applied in most environmental applications, Bayesian inversion with Gaussian prior. Even though the introduced methods are well studied in the inverse problem and compressed sensing communities, these methods have only been applied a few times to environmental problems. Despite some shortcomings in the evaluation, the presentation of their potential is welcome.

Code and figures are uploaded as supplementary material. Some instructions and comments are included to run the code. However, some scripts require input not shared by the authors.

In total, an interesting study that includes new modeling approaches in this setup. The experimental setup and the evaluation leave room for improvement. I recomment the publication after considering the following comments.

**2 Specific comments**

**Inverse problem**
The authors make a well structured introduction of the inverse problem and their solution methods. Though many readers will be familiar with some inverse problem, the methods used are rather unknown in the atmospheric research community. The introduction is therefore instructive. However, underdetermination is only one characteristic difficulty of inverse problems. Another typical property not discussed in detail in the manuscript is the sensitivity of the estimate to (noisy) data. I recommend to mention this property in the abstract (line 3) and include a small discussion in Sect. 2.1.

It could also be instructive to show a row of the sensing matrix as a footprint (as an addition to Fig. 2), though I assume that the currently shown matrix is only for illustration purposes and to small to hold a meaningful footprint.

**Sparse reconstruction**
Bayesian inversion with Gaussian prior is commonly applied to inverse problems in environmental studies. It is therefore the correct method to compare against. The method introduced by the authors, sparse reconstruction, has only been applied in atmospheric studies a few times. References are provided. A relation between least squares, Bayesian inversion with Gaussian prior and sparse reconstruction is nicely presented, but there are a few points to address:
The Bayesian inversion approach calculates a posterior distribution. Often, this posterior distribution is assessed by its maximum a posteriori solution as a best estimate and the (co-)variances as uncertainties. In the Gaussian case the interpretation of the posterior distribution is completely described by these parameters. Variational methods, Eq. (8), mainly focus on the best estimate by assuming certain properties via the penalty function, e.g. smoothness or sparsity. The penalty functions $R(\cdot)$ can be chosen more freely. Particularly, $C_2$ can be more general than a correlation matrix (cp. line 112). Some choices may be difficult to formulate as a Bayesian prior or the analysis of the posterior may become too difficult. However, the Bayesian equivalent to sparse reconstruction uses a Laplacian prior. This being said, I cannot completely agree with some formulations in the text, e.g. 'sparse reconstruction [...] does not require a prior emission field' (line 5). In this sense, neither does $l_2$-regularization. My feeling is that both inverse modeling interpretations are mixed and I recommend to review the text for imprecise formulations.

As a side note: I think the $l_0$-norm and the $l_p$-norm should also be defined for readers new to the topic (lines 133 and 167). Actually, I found the $l_0$-norm in Table 1, but the reader is not guided there in the first reading.

**Implementation of the methods**
The authors reduce the dimensionality of the optimization problems by removing insensitive grid cells and setting their values to zero. While a physical interpretation of insensitive grid cells is given, an explanation why this preprocessing step is justified is

missing (around line 247). In the standard domain the emissions on insensitive grid cells should be estimated as zero for both approaches. However, in the wavelet domain, I suppose that these emissions could (slightly) deviate as they may be parameterized by some wavelets with nonzero coefficients. Are the wavelets created prior to the removal or on the reduced system? In the latter case, the removal may influence the wavelet setup.

**Experimental setup**
During the first reading, I was not aware that most scenarios, e.g. Sect. 4.2, are evaluated without any noise on the measurements. The setup in each experiment should be clearly pointed out. Also, I am not sure which wind fields are used in the scenarios. Are they artificially generated? Is the wind uniform? I found some incomplete information in the Appendix.

In general, the setup in all experiments is idealized. It seems that most scenarios are evaluated with noiseless data. In the noisy case only measurement noise is included. Transport errors, uncertainties from the background and temporal variability of emissions are not considered. Also, the wind variation may be less ideal, assuming they are artifical, in an applied scenario. Aren't the results overly optimistic for any application?

In a theoretical study some simplification may be reasonable, but to me the methods lack a realistic test. For example, experiment 4.2.3 aims at discovering emissions not included in a good prior, e.g. an inventory. Small deviations from a good prior emission field should only produce small deviations in the observed enhancements. Are these detectable in the noisy case (cp. Sect. 4.2.5), particularly when including other sources of uncertainty (transport, background, etc.)?

Also, it seems that many results from the noiseless case do not transfer to the noisy evaluation, e.g. the wavelet approach performs less convincing.

In the noisy case, relative noise related to the measurement uncertainty and observed enhancements for the city of Indianapolis (not further evaluated) are considered. As measurement uncertainties are identical and expected trace gas enhancements can be calculated for each city, I wonder why a fixed SNR is used for all cities.

**Analysis**
While the discussion of compressed sensing is interesting in general, I am not convinced that this debate is that helpful. The options to design the sensing matrix in an optimal way for sparse methods are rather limited. As observations from the same location under similar wind conditions tend to have similar footprints, variety of measurement information can only come from changing wind conditions and different measurement locations. The authors find that even with full wind direction coverage the reconstruction criteria cannot be met, even in the noiseless case (also cp. App. Lines 443).

What is meant by Line 444: Some parameterization of the emissions as in the wavelet

approach may create different dependencies, but the footprints will always create some structure to the sensing matrix. Do you have an example?

Then, I doubt that the results of Fig. 5 should be interpreted with respect to the coherence, since the coherence measures the maximum similarity between columns. An identity matrix extended by the copy of one column has a terrible coherence of 1, but is the perfect sensing matrix for all but the two underdetermined parameters. Therefore, the coherence may be a measure for the maximum error - I do not know - but the results in Fig. 5 are probably better explained by a sensitivity analysis.

An explanation for the increase of the relative error of regularized least squares is missing. I also wonder why the results for sparse reconstruction in the wavelet domain is not included. Does it show similar behaviour to SR?

**Results**
In Fig. 4 it is a bit surprising to see that regularized least squares produce a somewhat sparse emission field, particularly in the outskirts. To me it looks as oscillations (negative emissions are not shown) originating from a regularization parameter $\lambda$ (Eq. 8) that is too small. This would also explain the large number of negative emissions. Many classical parameter choice rules reduce the model-data mismatch to a factor greater than 1 times noise level, before instabilities are introduced to the estimate. The same issue may apply to the sparse reconstruction approaches. Though the sparsity is increased with sparse reconstruction, the solution is not really sparse (only 10% or 20%) as discussed in Table 2. As it seems that this scenario considers noiseless data, forcing equality between model and data may be too strict or tolerances in the optimization routine could be fine tuned. In the noisy case, the authors also seem to observe overfitting (line 361).

It would be interesting to see the $l_2$-errors for the reconstructions in Fig. 4.

In general, it is also a bit surprising to see relative errors much larger than 1, e.g. in Fig. 8, for a stable inversion. What is the interpretation?

In Fig. 8, scenario (b) uses the highest emission resolution with varying number of observations. Shouldn't the results from scenario (a) at $1 \ km \times km$ line up at $\frac{m}{n} = 0.75$ in panel (b)? Maybe, I missunderstand.

**Code**
Thanks for including the code. Useful comments and instructions are provided. Inputs, except the inventories, are available for download. Maybe pseudo-inventories (with a warning in the code) could be created to make all codes executable. Overall, great effort to make the programming approaches available.

**3  Technical corrections**

Line 155: '... make good estimates of ...', estimation is the process of making an estimate

Line 255 and others: 'good compressible' and 'not good compressible', a better formulation should be found, e.g. 'compressible' and 'incompressible' (define what is meant by incompressible)

Line 234: 'sensing matrix matrix A', delete one 'matrix'

Line 243: '... sensitivity is beneath a certain threshold', 'below' works better

Line 332: '... descent ...', should be 'decent'

The code is more clear if variable x_l2 is used in the $l_2$-case in file *optimizeL2_ noise.m*.

---

## Author Comment (AC1)

Dear reviewer 1,

We are very grateful for your comments and suggestions, which have helped to improve our manuscript significantly. We have revised the manuscript accordingly, and the changes can be found in the track-changes file. The following is a point to point response to your comments and suggestions. Corresponding changes in the manuscript are also made available below at the appropriate places, if applicable.

Sincerely,

Benjamin Zanger and Jia Chen on behalf of all co-authors.
* * *
**Anonymous Reviewer #1:**

**Line 57: What is a 3rd-level wavelet transform? If emissions on a N x N grid are subjected to Haar transform, one gets a transform hierarchy of log2(N) levels. In a 3rd-level transform, do you ignore all levels finer than 3 (which would give a very rough emission field) or do you only keep level 3?**

The first level of the (Haar) wavelet transform decomposes the emission grid into high frequency components and low frequency components. The second level then acts on the low frequency part of the first level and decomposes it into high frequency components and low frequency components. A $3^{rd}$ level wavelet transform therefore consists of the high frequencies of the first 3 levels and the remaining low frequency part of the $3^{rd}$ level. For visualization, we refer to Fig. 1.1 in Mallat (1999) (a free version of the first 3 chapters can be found here: https://www.di.ens.fr/~mallat/papiers/WaveletTourChap1-2-3.pdf).

Thus, we do not ignore any levels, we only stop decomposing the low frequency part further.

We added a reference to Mallat (1999) in the text and referred to Fig. 1.1 for a visualization of a $3^{rd}$ level wavelet transform:

| Lines 183 ff: | *For an introduction to wavelets, we refer to Graps (1995) and Mallat (1999). For the visualization of a $3^{rd}$ level DWT, see Fig. 1.1 in Mallat (1999).* |
| --- | --- |

**Line 390: "SR is good at localization and SR-in-wavelet-domain is not robust to noise". I see that empirically from the results, but can you explain why? Does it have anything to do with the 3rd-level transform? Or Haar wavelets, which are oscillatory? Or because you do not impose non-negativity on the estimated emissions?**

Thank you for this question!

The reason why SR works well for localizing sources is as follows:

1. SR prefers sparse over smooth solutions. For the spatial domain, this means that preferably a single strong emitter is picked, while LS, which prefers smooth solutions, prefers to pick multiple small emitters to account for emissions. Therefore, SR is more capable of detecting point sources.
2. Compressed sensing (CS) property guarantees us the right answer for the emission location of the highest emitters. Even though the compressed sensing condition is not satisfied all the time, we still can have a "high probability" for the guarantees of CS for emission distributions which are similar/close to the true emissions.

Secondly, let us elaborate on the robustness of SR DWT towards noise. Please note, that the coefficients of the wavelet transformation have a huge difference in contribution to the measurement. To show this, we plotted the sensitivities of the measurement to the coefficients, given by the sum of the contributions of a coefficient i to each measurement j,

$$\sum_j \alpha_{ji}$$
,

where $\alpha_{ji}$ is the entry of the j-th row and i-th column of the sensing matrix:

[Figure]

[Figure]

While for the sensing matrix in the spatial domain (left side), those sensitivities do not differ too much in magnitude (except for the right outermost sensitivities), the sensitivities of the wavelet sensing matrix possess much higher differences. Furthermore, the large sensitivities belong to coefficients of large spatial areas, while the low sensitivities belong to small spatial areas.

This huge difference in sensitivity values is not ideal for fulfilling CS conditions properties, especially for CS condition properties needed for the noisy case, which are more strict than those in the noiseless case.

Furthermore, coefficients which have a lower contribution to the measurement are also more prone to noise, so that in the noisy case there will be a higher estimation error in those coefficients.

We also want to highlight Ray's approach (Ray et al., 2015) to make the SR DWT more robust to noise. In their paper, they used prior knowledge so that all estimates are in the same order of magnitude; hence bringing all sensitivities to the same order of magnitude as well. They showed that this improves the performance of the estimation.

However, we chose not to do this, because the prior knowledge is not accessible for unknown emitters.

One could try other wavelets (other levels), but according to the explanation we provided above, this would not solve the problem. We think that other domain transformations than wavelets could improve the situation. However, it is not clear which domain transformations are better suited, since a transformation always changes sparsity of the signal as well as the properties of the resulting sensing matrix. It would be interesting to perform a systematic study of other domain transforms in a future study.

**In Sec 4.2.2, 4.2.3 and 4.2.4, what are the wind coverage in the cities?**

In Sec. 4.2.2, 4.2.3 and 4.2.4, the same wind fields are used. These wind fields have a wind coverage of about 143°. We added this information to the main text:

| Line 246: | *In our work, we use artificial wind data. These wind fields used, except in Sec. 4.2.1, possess a wind coverage of ~143°. All of the footprints generated by those wind fields are available at* [https://doi.org/10.5281/zenodo.5901298](https://doi.org/10.5281/zenodo.5901298) *(Zanger et al., 2022a).* |
|---|---|

Please keep in mind that this number is not strictly comparable to the wind coverage used in Sec 4.2.1, because in Sec 4.2.1 a static Gaussian plume model is used, while the other sections use a non-static Gaussian plume model. Please also see the details why we are using a static model for Sec 4.2.1 in the Appendix B:

| Lines 460 ff: | *For Sec. 4.2.1, we use a static Gaussian plume model to generate footprints for different wind coverages. This allows us to ignore additional effects from the dynamical case. For example, in the dynamic case the change in the wind direction produces spiral looking footprints. These footprints have a different shape compared to the footprints generated using a static Gaussian plume model . By using a static model, the footprints are more predictable and systematic, which makes the results of different wind coverages more comparable to each other.* |
|---|---|

**Are the wind velocities the same or are the footprints stretched to cover the cities which are of different sizes?**

The footprints are stretched and the velocities change. Please see Appendix B:

| Lines 457 ff: | *The footprints are then scaled to the domain size of the city. Because of the scaling, the wind speed and diffusion changes for each city. Nevertheless, we think that this approach makes the reconstruction performances for different cities more comparable than using different footprints for every city.* |
| --- | --- |

**Fig 7 and Fig. D1: The x-axis has bins called "0 - 0.4%" ; the label on the x-axis is "Highest emission in inventory". What does it mean? Do you rank-order the gridded emissions, and bin them in unequal bins? How were the bin cutoffs decided? Should that be cast in terms of percentiles of the gridded emissions? Or is it the percentiles of their ranks? Some explanation of what this axis is, in the text or captions, would be helpful**

Thank you for this comment. We added an explanation in the text:

| Lines 331 ff: | *The x-axes show bins of emission grids ranked by their emission strength. For example, the first bin (0 - 0.4%) contains the first 0.4% highest emission grids while the last bin in the lower panels of Fig. 7 contains emission grids which are among the 36.2 - 53.2% highest emission grids. Since the emissions are highly compressible (first highest emissions have a huge contribution to the total emissions) we use a logarithmic scale for the bins.* |
| --- | --- |

**The English in Sec 3, 4 and 5 needs to be improved. [...]**

We contacted the language center at TUM and a lecturer there (native speaker) has gone through our text. The language of our paper has been improved. Please see the track-change version to see our changes.

**The is a lot of use of the emissions being "good compressible" or "well compressible". I think one could simply use "compressible"[...] Also, in the text and tables, one sees emissions "not good compressible" or "not well compressible". What about non-compressible?**

Thank you for the suggestions. We adapted the wording to "good compressible" and "not good compressible". We think that "non-compressible" would be too strict, since in fact those emissions are still compressible ($\sigma_{10\%}$ for Paris is 0.217).

**The is also use of the terms "amount of measurements" and "less measurements". Measurements in this paper are counts. What about using "number of measurements" and "fewer measurements"?**

Thank you for this comment, we adapted it in the paper.

**Line 332: The authors talk about estimating the measurement counts needed to achieve "descent results". That is rather casual. What about "acceptable results"? Also, what would constitute an "acceptable result"? Please clarify**

Thank you for this comment. We wanted to find a general relation between the number of measurements and the accuracy. We changed the wording accordingly:

| Lines 352 ff: | *However, the question of how many measurements m are needed to produce results of a certain accuracy remains unanswered.* |
|---|---|

**Fig 7: The caption mentions "how many percents ...." This is hard to decipher. What about explaining the figure in the text where there is no shortage of space?**

Thank you for this suggestion. We removed the explanation from the caption in Fig. 7 and improved the explanation of the qualitative and quantitative measure in the text (please see line 330 ff).
* * *
**References:**

Aster, R.C., Borchers, B. and Thurber, C.H., 2018. *Parameter estimation and inverse problems*. Elsevier.

Tarantola, A., 2005. *Inverse problem theory and methods for model parameter estimation*. Society for Industrial and Applied Mathematics.

Ray, J., Lee, J., Yadav, V., Lefantzi, S., Michalak, A., and Bloemen Waanders, B. v., 2015. *A sparse reconstruction method for the estimation of multi-resolution emission fields via atmospheric inversion*. Geoscientific Model Development, 8, 1259–1273. https://doi.org/10.5194/gmd-8-1259-2015.

Ray, J., Yadav, V., Michalak, A. M., van Bloemen Waanders, B., and McKenna, S. A.: A multiresolution spatial parameterization for the estimation of fossil-fuel carbon dioxide emissions via atmospheric inversions, Geosci. Model Dev., 7, 1901–1918, https://doi.org/10.5194/gmd-7-1901-2014, 2014.

Hase, N., Miller, S. M., Maaß, P., Notholt, J., Palm, M., and Warneke, T.: Atmospheric inverse modeling via sparse reconstruction, Geosci. Model Dev., 10, 3695–3713, https://doi.org/10.5194/gmd-10-3695-2017, 2017.

Jones, T.S., Franklin, J.E., Chen, J., Dietrich, F., Hajny, K.D., Paetzold, J.C., Wenzel, A., Gately, C., Gottlieb, E., Parker, H. and Dubey, M., 2021. Assessing urban methane emissions using column-observing portable Fourier transform infrared (FTIR) spectrometers and a novel Bayesian inversion framework. *Atmospheric Chemistry and Physics*, *21*(17), pp.13131-13147.

Lauvaux, T., Miles, N. L., Deng, A., Richardson, S. J., Cambaliza, M. O., Davis, K. J., Gaudet, B., Gurney, K. R., Huang, J., O'Keefe, D., Song, Y., Karion, A., Oda, T., Patarasuk, R., Razlivanov, I., Sarmiento, D., Shepson, P., Sweeney, C., Turnbull, J., & Wu, K. (2016). High-resolution atmospheric inversion of urban $CO_2$ emissions during the dormant season of the Indianapolis Flux Experiment (INFLUX). *Journal of geophysical research. Atmospheres : JGR*, *121*(10), 5213–5236. https://doi.org/10.1002/2015JD024473.

Mallat, S., 1999. *A wavelet tour of signal processing*. Elsevier.

---

## Author Comment (AC2)

Dear reviewer 2,

We are very grateful for your comments and suggestions, which have helped to improve our manuscript significantly. We have revised the manuscript accordingly, and the changes can be found in the track-changes file. The following is a point to point response to your comments and suggestions. Corresponding changes in the manuscript are also made available below at the appropriate places, if applicable.

Sincerely,

Benjamin Zanger and Jia Chen on behalf of all co-authors.
* * *
**Anonymous Reviewer #2:**

**The authors make a well structured introduction of the inverse problem and their solution methods. Though many readers will be familiar with some inverse problem, the methods used are rather unknown in the atmospheric research community. The introduction is therefore instructive. However, underdetermination is only one characteristic diculty of inverse problems. Another typical property not discussed in detail in the manuscript is the sensitivity of the estimate to (noisy) data. I recommend to mention this property in the abstract (line 3) and include a small discussion in Sect. 2.1.**

Thank you for mentioning this. We changed the abstract to the following:

| Lines 2 ff: | *Solving this inverse problem is challenging because the system of equations often has no unique solution and the solution can be sensitive to noise.* |
|---|---|

We also added more details about ill-posed problems and the features we deal with in this paper in Sec. 2.1:

| Lines 79 ff: | *Often, however, such inverse problems are ill-posed, as in the cases we deal with in this paper. In such cases, no or no unique solution exists or the solution does not depend smoothly on the data, therefore, being sensitive to noise. For ill-posed inverse problems, the least squares estimation without regularization does not provide a useful reconstruction technique. For a more detailed discussion of ill posed problems we refer to chapter 3 of Nakamura and Potthast (2015).* |
|---|---|

**It could also be instructive to show a row of the sensing matrix as a footprint (as an addition to Fig. 2), though I assume that the currently shown matrix is only for illustration purposes and too small to hold a meaningful footprint.**

Thank you for this suggestion. Indeed, the matrix in Fig. 2 is only for illustration purposes. We are showing two footprints in Appendix B. We slightly adapted the text to make the reader aware that the footprints are shown in the Appendix:

| Lines 241: | *The rows in A contain vectorized footprints, which determine the sensitivity of the measurements to the GHG fluxes of different emission grid cells in the domain. These footprints are determined by backward transport  models, such as the Stochastic Time-Inverted Lagrangian Transport (STILT) model (Lin et al., 2003; Gerbig et al., 2003). In this paper, a simplified linear model, the Gaussian plume model, is used as the transport model. This approach allows varying parameters within the transport model without the computationally costly calculations of the STILT model. In Appendix B, we explain how the Gaussian plume footprints are created and show two footprints, one calculated by STILT, the other by the Gaussian plume model (Fig. B1).* |
|---|---|

**[...]. However, the Bayesian equivalent to sparse reconstruction uses a Laplacian prior. This being said, I cannot completely agree with some formulations in the text, e.g. 'sparse reconstruction [...] does not require a prior emission field' (line 5). In this sense, neither does l2-regularization. My feeling is that both inverse modeling interpretations are mixed and I recommend to review the text for imprecise formulations.**

Thank you for this comment. Indeed, our formulation has been imprecise. Both SR (Laplacian prior) and LS (Gaussian prior) can use emission priors $x_a$, but do not necessarily need them.
However, the assumption of a Laplacian distribution of emissions, or any sparser distribution gives precise results with bounded error if compressed sensing conditions hold. Also see the description we give in the introduction:

| Lines 30 ff: | *These [SR] methods determine the critical emission grid cells and adjust the emissions of only those cells until the model best matches the observations. All other grid cells are set to zero.  Once conditions of compressed sensing (CS) are fulfilled, SR methods are guaranteed to determine the best possible emission grid fields and provide a good estimation of their emissions.* |
|---|---|

We also changed the abstract accordingly to clarify that sparse reconstruction "can achieve reasonable estimations without using a prior emission field". Please see the following changes we did in the abstract:

| Lines 5 ff: | *In our work, we investigate sparse reconstruction (SR), an alternative reconstruction method that  can achieve reasonable estimations without using a prior emission field by making the assumption that the emission field is sparse.* |
|---|---|

We changed the wording of the text in the paper and added some more context to the paper. The changes are listed below:

| Lines 114 ff: | *For such a regularization function, the regularization scheme is known as Tikhonov regularization or also ridge regression (Golub et al., 1999). While in Bayesian inversion it is most often assumed that a prior value $x_A$ is known, in regularization $x_A$ is often unknown and assumed to be 0.* |
|---|---|

| Lines 121 ff: | *The equivalent of the Lasso in Bayesian inversion is the assumption of a Laplacian distributed prior.* |
|---|---|

**As a side note: I think the l0-norm and the lp-norm should also be defined for readers new to the topic (lines 133 and 167). Actually, I found the l0-norm in Table 1, but the reader is not guided there in the rst reading.**

Thank you for this comment. We added the definition of the $l_0$ norm and the $l_p$ norms to Table 1:

**Norms**

| | | |
|---|---|---|
| $l_p$ norm | $\left(\sum_i x_i^p\right)^{\frac{1}{p}}$ | $l_p$ norm for vectors $x \in \mathbb{R}^n$. |
| $l_0$ norm | $|\{j|x_j \neq 0\}|$ | Number of non zero elements in $x$. |

We also added a reference to this table in section 2.4:

| Lines 134: | *An overview of the symbols, norms, and measures used is given in Table 1.* |
|---|---|

**The authors reduce the dimensionality of the optimization problems by removing insensitive grid cells and setting their values to zero. While a physical interpretation of insensitive grid cells is given, an explanation why this preprocessing step is justied is missing (around line 247). In the standard domain the emissions on insensitive grid cells should be estimated as zero for both approaches.**

Mathematically, a low value in the sensing matrix will only result in a small contribution to the measurement. Assume the parameter $a_{ji}$ is an order of magnitude smaller compared to the other parameters in the j$^{th}$ row of the matrix A. The j$^{th}$ measurement is given by:

$y_j = a_{j1}x_1 + \cdots + a_{ji-1}x_{i-1} + a_{ji}x_i + a_{ji+1}x_{i+1} + \cdots + a_{jn}x_n$

Assuming that $\alpha_{ji}x_i << y_j \quad \forall j$, which we ensure by

$$\sum_j \alpha_{ji} < \gamma$$

and setting a small enough threshold $\gamma$.Then:

$y_j \approx a_{j1}x_1 + \cdots + a_{ji-1}x_{i-1} + a_{ji+1}x_{i+1} + \cdots + a_j n x_n \quad \forall j$

Therefore, calculating $x_1, \ldots, x_{i-1}, x_{i+1}, \ldots, x_n$ from these perturbed measurements is similar as reconstructing those emission grid cells with noisy measurements with a very high SNR (where the SNR depends on the threshold $\gamma$ we choose).

Indeed, without reducing the dimensionality of the optimization problems, both regularized LS and SR would estimate those emissions as zero, or close to zero. Physically, however, it does not make much sense to reconstruct emissions grid cells which we are not measuring. We remove these insensitive grid cells to reduce the dimensionality of the problem, also because it would reduce the computational effort and improve numerical stability, since it improves the conditioning of matrix A.

We also add an explanation to the main text:

| Lines 253 ff: | *For Fig. 2 this would be the first column of the sensing matrix, where all values are 0. This introduces some small perturbation between y and Ax, so that Ax $\approx$ y. However, since we choose our threshold small enough, this is not reflected in the solution but improves the conditioning of A and reduces the computational effort.* |
|---|---|
| | *From a physical point of view, these removed emission grid cells are not situated upwind of the measurement stations and, therefore, cannot be well reconstructed using the measurements.* |

However, if the reviewer suggests that we should not use this threshold method because of mathematical consistency, we can also remove this part in the paper.

**However, in the wavelet domain, suppose that these emissions could (slightly) deviate as they may be parameterized by some wavelets with nonzero coefficients. Are the wavelets created prior to the removal or on the reduced system? In the latter case, the removal may influence the wavelet setup.**

Thank you for this comment. In the wavelet domain we perform this step after the wavelet transform, so where the sensing matrix is given by $\bar{A} = W^{-1}A$. Therefore, the wavelets

are created prior to the removal and the removal will not influence the wavelet setup. We added this explanation into the paper:

| Lines 257 ff: | *For the wavelet domain, we perform this step on the wavelet transformed sensing matrix. Therefore, wavelet coefficients which are weakly sensed and below the threshold, are removed.* |
|---|---|

**During the first reading, I was not aware that most scenarios, e.g. Sect. 4.2, are evaluated without any noise on the measurements. The setup in each experiment should be clearly pointed out.**

Thank you for mentioning this. We changed the text, so that it is clear which sections are with and without noise:

| Lines 278 ff: | *We first analyze those properties in the noiseless case (Sec 4.2.1 to 4.2.4) and then show examples including measurement noise (Sec. 4.2.5).* |
|---|---|

**Also, I am not sure which wind fields are used in the scenarios. Are they artificially generated? Is the wind uniform? I found some incomplete information in the Appendix.**

The wind fields we use are artificially generated and have a wind coverage of ~143°. All of the footprints generated by these windfields are also available online at *https://doi.org/10.5281/zenodo.5901298*. We have added this information to the main text of the paper:

| Lines 246 ff: | *In our work, we use artificial wind data. These wind fields used, except in Sec. 4.2.1, possess a wind coverage of ~143°. All of the footprints generated by those wind fields are available at https://doi.org/10.5281/zenodo.5901298 (Zanger et al., 2022a).* |
|---|---|

Section 4.2.1 uses different wind fields than the other sections, generated by a static Gaussian plume model. The explanation is found in Appendix A:

| Lines 460 ff: | *For Sec. 4.2.1, we use a static Gaussian plume model to generate footprints for different wind coverages. This allows us to ignore additional effects from the dynamical case. For example, in the dynamic case the change in the wind direction produces spiral looking footprints. These footprints have a different shape compared to the footprints generated using a static Gaussian plume model . By using a* |
|---|---|

|  | *static model, the footprints are more predictable and systematic, which makes the results of different wind coverages more comparable to each other.* |

**In general, the setup in all experiments is idealized. It seems that most scenarios are evaluated with noiseless data. In the noisy case only measurement noise is included. Transport errors, uncertainties from the background and temporal variability of emissions are not considered. Also, the wind variation may be less ideal, assuming they are artificial, in an applied scenario. Aren't the results overly optimistic for any application?**

In reality, the SNR may be lower. But compared to existing literature, the SNR we assume is closer to reality (In Ray et al., 2015: 0.1 ppmv, >35 dB, Hase et al., 2017: mention that realistic noise is used, without stating details). Our paper could serve as a first step into the direction of applying SR to urban emissions and we want to demonstrate the capability of our new method here. Demonstrating these potentials is easier in settings with a low amount of noise. As a future step, we plan to apply our method to real data (e.g. from MUCCnet, https://amt.copernicus.org/articles/14/1111/2021/) and consider the uncertainties from transport, background, etc.

The transport error, background error, temporal variability of emissions are hard to estimate and vary from case to case. We, nevertheless, believe that our noise assumptions are generalizable. To show this, we plotted the rel l2 error of SR, SR DWT, and LS for different SNRs using the emission map of Munich and $m/n \approx 75\%$. We run each reconstruction method on 5 different noise realizations of the same strength. The Figure is depicted below:

[Figure]

As you can see, the 20 dB case is qualitatively not significantly different from cases with lower SNRs (till 10-15 dB), where SR always performs much better and SR DWT performs better than LS. There are slight changes in the rel. $l_2$ error with changing SNR, but those are

approximate linear and most probably are not due qualitative differences in the reconstruction. Huge changes can be found by increasing the SNR significantly for SR DWT (>30 dB), which is more likely due to qualitative changes (the SNR is probably high enough that the sensitivity differences to the different coefficients are not important anymore, please see answer to the second question of reviewer #1). We also added this Figure and the information into our paper:

| Lines 376 ff: | *Figure 9 depicts how the rel. $l_2$ error changes for different SNRs in the Munich simulation. The figure shows that the results we obtain for an SNR of 20 dB should not differ significantly for slightly lower or higher SNRs, between 15 dB and 25 dB. Therefore, we believe that despite our noise assumptions, which do not reflect real world scenarios adequately, our results are qualitatively valid.* |
|---|---|

**In a theoretical study some simplification may be reasonable, but to me the methods lack a realistic test. For example, experiment 4.2.3 aims at discovering emissions not included in a good prior, e.g. an inventory. Small deviations from a good prior emission field should only produce small deviations in the observed enhancements. Are these detectable in the noisy case (cp. Sect. 4.2.5), particularly when including other sources of uncertainty (transport, background, etc.)?**

We added an analysis of detecting unknown emitters for the noisy case in section 4.2.5 (see Figure 11, please compare it to Figure 7 for noiseless case), using the same noise as for the other noisy cases (SNR = 20 dB).

SR in the spatial domain is still sensitive to the largest emitters (0-0.4%), while SR DWT is more sensitive to smaller emitters (0.4% - 6.2%) . We also included the statement in the main text:

| Lines 398 ff: | *SR is the only method to identify the 0.4 % highest emitters (qualitative measure) and estimates their emissions with a significantly lower rel. error of 2 % (quantitative measure). Slightly smaller emitters are better estimated by the SR in the wavelet domain. For even lower emitters, there is no clear winner of which method estimates them best. Compared to the noiseless case, the estimation errors are in general larger and SR in the wavelet domain does not achieve as promising results as in the noiseless case. SR in the spatial domain remains sensitive to the largest emitters in the noisy case. These results for Paris are found in the supplement.* |
|---|---|

The results for Paris are given in the supplement. For Paris's emissions, which are not well compressible, we see that SR is not sensitive to high emitters and SR DWT performs

similarly to LS (see supplement figures "finding_unknown_emitters_paris/Qualitative_measure.pdf" and "finding_unknown_emitters_paris/Quantitative_measure.pdf").

We did not include other sources of uncertainty. For the explanation, please see the answer to the previous question.

In general, the reviewer is right. It is indeed challenging to detect small unknown sources in noisy cases. Nevertheless, our simulation with noise shows that it is possible. In the future studies, we will use real data (e.g. data from our MUCCnet), and also conduct further investigations and sensitivity studies to answer the question of what is the smallest unknown emitter that we can detect under different noise conditions.

**Also, it seems that many results from the noiseless case do not transfer to the noisy evaluation, e.g. the wavelet approach performs less convincing.**

Yes, the reviewer is right, especially for the sparse reconstruction in the wavelet domain, the results from the noiseless cases do not directly transfer to the noisy cases. This is because SR in the wavelet domain does not seem to be robust against noise (please see the answer to the second question of reviewer #1).

Nevertheless, for the SR spatial domain most properties still transfer from noiseless cases to noisy cases.

**In the noisy case, relative noise related to the measurement uncertainty and observed enhancements for the city of Indianapolis (not further evaluated) are considered. As measurement uncertainties are identical and expected trace gas enhancements can be calculated for each city, I wonder why a fixed SNR is used for all cities.**

Thank you for this question. Even if we scale the SNR with the expected enhancements, the resulting SNR for different cities would be very similar. In the following, we scale the SNR with the mean emission flux of each city, using London as a reference (20 dB).

| | London | Paris | Berlin | Hamburg | Vienna | Munich |
|---|---|---|---|---|---|---|
| **SNR** | 20.0 dB | 18.9 dB | 19.8 dB | 19.0 dB | 20.6 dB | 20.5 dB |

As can be seen in the table, these SNRs for different cities are quite similar, where Paris has the lowest SNR with 18.9 dB and Vienna the highest with 20.6 dB. Therefore, we believe that using a fixed SNR for all cities is reasonable and the uncertainty introduced is insignificant compared to other uncertainties.

**While the discussion of compressed sensing is interesting in general, I am not convinced that this debate is that helpful. The options to design the sensing matrix in an optimal way for sparse methods are rather limited. As observations from the same location under similar wind conditions tend to have similar footprints, variety of measurement information can only come from changing wind conditions and different measurement locations. The authors mentioned that even with full wind direction coverage the reconstruction criteria cannot be met, even in the noiseless case (also cp. App. Lines 443).**

Compressed sensing is a novel approach, whose potential has not been thoroughly investigated for atmospheric inversion, especially for cities. In general, it is suited for recovering sparse emissions (Ray et al. 2015), and most cities' emissions are sparse.

Compressive sensing is theoretically more precise compared to Bayesian inversion in terms of localization and quantification of the emission strength, if the condition is fulfilled and the signals are sparse or compressible.  To examine the conditions, we chose coherence, since it is inexpensive to calculate, but it only provides a very loose bound. There are other criteria such as the RIP, which can be further explored in future studies. If the reconstruction criteria are fulfilled, we have a mathematical guarantee that the inversion can be performed. However, in real applications one can expect to get good results even though these properties are not fulfilled in the general case (for more information on this, please see http://www.irisa.fr/metiss/gribonval/Talks/GribonvalWorkshopSparsityInversePbCambridgeDec2008.pdf).

Because of these reasons, even if the reconstruction criteria cannot be met, compressed sensing can provide helpful information for designing purposes and one can still expect to achieve solutions that have better accuracy in terms of localizing and quantifying large emitters. This makes, in our point of view, the theory of compressed sensing interesting for the geoscientific community.

In the future, we will also try to match the conditions better by considering the following aspects in following studies, to improve the design of the sensor network and inversion framework:
- Changing the domain for reconstruction of the emissions
- Optimizing the sensor location, adding more sensors
- Move the sensors during the time of observation, so that best results are achieved
- Observing over a long time period

**What is meant by Line 444: Some parameterization of the emissions as in the wavelet approach may create different dependencies, but the footprints will always create some structure to the sensing matrix. Do you have an example?**

We are not quite sure if line 444 is the text you are referring to. In line 444, we have written "Nevertheless, A can be useful if it satisfies the CS conditions for some specific emission distributions of interest." Nevertheless, we try to answer the question as precisely as possible.

The footprints of a city are not randomly chosen, but are created by a parameterized process, where the parameters include for example wind. Because of this common process, the footprints don't look randomly distributed, but have some kind of "structure". As an example, we show two footprints of Munich from different dates:

[Figure]

[Figure]

Since the sensing matrix consists of the footprints, also the sensing matrix has a certain structure. What is important for compressed sensing is that this structure should possess CS conditions with high probabilities. You can assign a probability to the structure of the footprint by taking the probabilities of input parameters (wind, diffusion, …) and passing them through the process.. What we are saying in line 444 (now 470) is

| Lines 470 ff: | *Nevertheless, A can be useful if it satisfies the CS conditions for some specific emission distributions of interest.* |
|---|---|

As a reminder, let us look at the definition of the RIP in Eq. (10). There, the vector $x$ selects a $\mathbb{R}^{m \times s}$ submatrix of $A$ and checks how close these singular values are to 1 (for $\delta_s = 0$, they are all exactly 1). This means, the RIP checks for all s-sparse vectors of $x$ if RIP is satisfied. However, in reality emissions are not distributed randomly, but possess some structures. Therefore, it would be enough for us that the RIP is satisfied for the distributions of interests. Of course, we don't really know this distribution. Finding such distributions is currently progress of research and is e.g. useful for dimension reduction of the problem and (most often) used with Gaussian priors. However, in Appendix C we created s-sparse emissions of a city by having a higher probability that the emitters are located in the city center (in the middle of the reconstruction domain). As you can see from our reconstruction results, SR provides us with perfect reconstruction up to a large amount of emitters. Therefore, these emission distributions seem to satisfy the RIP (in the noiseless case).

Now to the Wavelet domain: A transformation changes the original structure of the sensing matrix to another one. However, some structure will remain in the sensing matrix. It would be ideal to find a transformation which changes the structure, so that CS conditions are satisfied with a higher probability and the emissions to be more compressible in this new domain. However, doing this is challenging. In our paper, we used the wavelet domain, since in its new

domain the emissions are sparser. But from our results, it is likely that RIP is satisfied with a lower probability for the noisy case.

**Then, I doubt that the results of Fig. 5 should be interpreted with respect to the coherence, since the coherence measures the maximum similarity between columns. An identity matrix extended by the copy of one column has a terrible coherence of 1, but is the perfect sensing matrix for all but the two underdetermined parameters. Therefore, the coherence may be a measure for the maximum error - I do not know - but the results in Fig. 5 are probably better explained by a sensitivity analysis.**

Indeed, the incoherence property provides a very loose bound for checking CS conditions. There are other criteria such as the RIP, which can be further explored in future studies. We want to emphasize in that section that by using CS properties, such as the incoherence property, to find better configurations of an experiment (demonstrated with the wind coverage), can increase the performance of SR, even if the CS properties are not satisfied. For an intuitive explanation of this, please see here:
http://www.irisa.fr/metiss/gribonval/Talks/GribonvalWorkshopSparsityInversePbCambridgeDec2008.pdf.

To the best of our knowledge, performing a global sensitivity analysis for SR, such as in LS using the sensitivity matrix, is still ongoing research and not straight forward. What is possible is a fixed sensitivity analysis, such as we have performed in Appendix E for the purpose of uncertainty quantification.

**An explanation for the increase of the relative error of regularized least squares is missing.**

Thank you for pointing this out. The increased error of regularized least squares is not a general phenomenon. It is case specific and for the example we showed in this paper we provide the following explanation. First, let us look at the emissions of Munich, with the colors denoting the emission strength in a unit of $\mu$ mol m$^{-2}$ s$^{-1}$:

[Figure]

Then please see the sensitivity analysis of the least squares for a wind coverage of 72° and a wind coverage of 360° (yellow means very sensitive, blue means not sensitive) :

[Figure]

As you can see, the measurements with a wind coverage of 72° give a much better reconstruction of the south east of the city, since this is where most of the wind comes from. For the 360° wind coverage, every grid cell is captured with a moderate sensitivity (see color scale), but by accident not the location of the highest emitter in Munich, which is located in the south east (compare with the highest emitter in the emission map in the Figure above). Therefore, in this special case we show in the paper the error tends to get larger for regularized least squares with increasing wind coverage, but the conclusion can not be generalized. We added this information to the appendix (see Appendix F):

| Lines 292 ff: | *A sensitivity analysis, given in Appendix F, reveals that error variations given by the different wind coverages are due to the shift of sensitivity to different parts of the domain, while the sum of the sensitivities do not change.* |
|---|---|

**I also wonder why the results for sparse reconstruction in the wavelet domain is not included. Does it show similar behaviour to SR?**

Thank you for this question. We checked the relationship between the coherence and wind coverage for the wavelet domain. In the wavelet domain the coherence also reduces with increasing wind coverage,

Therefore, we added the wavelet domain also in Fig. 5:

[Figure]

(a) Munich                (b) Paris

We also adapted the text in the paper:

| Lines 289 ff: | *Errors of the estimates for both SR in the spatial and wavelet domain decrease with an increasing wind coverage for Munich and Paris. In contrast, the LS estimates reveal no clear improvement for higher wind coverages, but its rel. error increases with high coverages for Munich, while for Paris the error initially decreases, until about a coverage of 150°, before increasing again. A sensitivity analysis, given in Appendix F, reveals that error variations given by the different wind coverages are due to the shift of sensitivity to different parts of the domain, while the sum of the sensitivities do not change.* |
| --- | --- |
| | *This major improvement of SR for higher wind coverage can be explained using the incoherence property from CS. The coherence parameter, given in Eq. (11), calculates the maximum similarity of how two emission grid cells are measured. For the spatial domain, by increasing the wind coverage, there is greater variety in the measurements and thus between the measurements of two different grid cells. In the wavelet domain, we determined the coherences at different wind coverages and found a similar behavior.* |

**In Fig. 4 it is a bit surprising to see that regularized least squares produce a somewhat sparse emission field, particularly in the outskirts.**

Yes, in Fig. 4 it seems as if regularized least squares would produce sparse emission fields, but those are actually oscillations (with negative emissions) and look sparse since we only plot positive emissions.

**To me it looks as oscillations (negative emissions are not shown) originating from a regularization parameter λ (Eq. 8) that is too small. This would also explain the large number of negative emissions. Many classical parameter choice rules reduce the model-data mismatch to a factor greater than 1 times noise level, before instabilities are introduced to the estimate. The same issue may apply to the sparse reconstruction approaches. [...]**

This scenario is in the noiseless case. For LS, we are using pseudo inverse (see line 206) and choose the tolerance level (corresponds to the lambda) high enough, so that it can not be the reason for the oscillation. For SR, we do not solve the regularization parameter lambda (see equation P1$\epsilon$), so the same issue does not apply to SR.

**As it seems that this scenario considers noiseless data, forcing equality between model and data may be too strict or tolerances in the optimization routine could be fine tuned. In the noisy case, the authors also seem to observe overtting (line 361).**

Yes, Fig. 4 considers $m/n \approx 0.75$ without noise. We are using a pseudoinverse with a high enough tolerance (corresponds to the lambda) so that machine precision errors, and others, do not influence the result.

We believe that the oscillating behavior in LS is due to the high resolution emission field to be estimated and not enough measurements available. This is known as the "Discrete Ill Posed problem". For an explanation please see chapter 3.5 of Aster et al. (2018) or chapter 3.3 of Tarantola (2005).

**Though the sparsity is increased with sparse reconstruction, the solution is not really sparse (only 10% or 20%) as discussed in Table 2.**

For our method, compressibility of the emissions is important, not necessarily the "sparsity" given as $\|x\|_0$ (see Table 1). We define compressible emissions by emissions which can be well approximated by sparse emissions (see Sec. 2.5 and Fig. 1). What is seen in Table 2 on the two leftmost columns are the approximation errors of the emissions if approximated by their 10% highest emitters. This is the best possible approximation of the emissions for a relative sparsity of 10% (Definition of this measure is given in Table 1).
As you see, for the DWT emissions, this approximation error is less. Hence, the emissions are more compressible.

**It would be interesting to see the l2-errors for the reconstructions in Fig. 4.**

The rel. l2 errors for Fig. 4 are actually given in Table 3. We added a reference to this table in the text:

| Lines 270 ff: | *Reconstruction results for Munich in the noiseless case using LS, SR, and SR in the wavelet domain are depicted in Fig. 4, and the reconstruction errors for the figure can be found in Table 3.* |
|---|---|

In case you are more interested in a map of the rel. l2 errors, they are given below. Those are actually not the elementwise rel. l2 errors, but divided by the l2 norm of the total emissions. More concretely:

$$\frac{|\hat{x}_i - x_i|}{\|x\|_2}$$

We believe that this is more useful than an elementwise rel. l2 error, since it depicts the contribution of the error of the pixels to the rel. l2 error.

[Figure]

We will put this as additional information into the supplement and to the code implementation, so that interested readers are able to look deeper into this.

**In general, it is also a bit surprising to see relative errors much larger than 1, e.g. in Fig. 8, for a stable inversion. What is the interpretation?**

Such relative errors larger than 1 are due to localization error of the emitters. This especially happens because our emission fields are of high resolutions. For the total emissions, there are no relative errors of 1 or larger.

Let us provide an example for clarification. Assume a single emitter on an emission map which emits 10 units of emissions (unit doesn't matter here):

[Figure]

Now assume a reconstruction, in which the emission strength is perfectly reconstructed but the location of the emitter is wrongly estimated (northeast of the real location):

[Figure]

In this case, the rel. $l_2$ error is $\sqrt{10^2 + 10^2}/10 = 1.414$, which is a rel. $l_2$ error larger than 1 To make this also clear to the reader, we added the sentence in the paper:

| Lines 213 ff: | *Because of the high spatial resolution of the emission fields reconstructed in this paper, rel. $l_2$ error values greater then 1 are possible. Those are because of spatial errors, where an emission is assigned to the wrong spatial location. The rel. total error disregards these spatial errors and only evaluates the difference of the total estimated emission to the true total emission of a city.* |
|---|---|

**In Fig. 8, scenario (b) uses the highest emission resolution with varying number of observations. Shouldn't the results from scenario (a) at 1 km × 1 km line up at m/n = 0.75 in panel (b)? Maybe, I missunderstand.**

Thank you for this comment. We assume you mean Fig. 9 and will answer the question for that figure. We did not  mention that Fig. 9, left, is run on multiple instances of the noise vector and added this information to the new version of the text and also in the figure description:

| Lines 382 ff: | *For this case we have run the reconstruction on 250 different noise vectors and show the mean estimation result.* |

For Fig. 9 right, we only used a single noise vector for every point we plotted. Therefore, these plots are not strictly comparable. But you are right, the expectation value from scenario (a) at 1 km × 1 km should be the same as m/n = 0.75 in panel (b).

**Thanks for including the code. Useful comments and instructions are provided. Inputs, except the inventories, are available for download. Maybe pseudo-inventories (with a warning in the code) could be created to make all codes executable. Overall, great effort to make the programming approaches available.**

Thank you for this comment. We added pseudo inventories to the code.

**Line 155: '... make good estimates of ...', estimation is the process of making an estimate**

Thank you for the remark, we changed that in the paper.

**Line 255 and others: 'good compressible' and 'not good compressible', a better formulation should be found, e.g. 'compressible' and 'incompressible' (define what is meant by incompressible)**

We changed the wording to "highly compressible" and "slightly compressible" throughout the paper.

**Line 234: 'sensing matrix matrix A', delete one 'matrix'**

Thank you for the remark, we changed that in the paper.

**Line 243: '... sensitivity is beneath a certain threshold', 'below' works better**

Thank you for the remark, we changed that in the paper.

**Line 332: '... descent ...', should be 'decent'**

Thank you for the remark. We changed the word to "accurate".

**The code is more clear if variable x_l2 is used in the l2-case in le optimizeL2_noise.m.**

Thank you for this suggestion, we changed this in the code.
* * *
**References:**

Aster, R.C., Borchers, B. and Thurber, C.H., 2018. *Parameter estimation and inverse problems*. Elsevier.

Tarantola, A., 2005. *Inverse problem theory and methods for model parameter estimation*. Society for Industrial and Applied Mathematics.

Ray, J., Lee, J., Yadav, V., Lefantzi, S., Michalak, A., and Bloemen Waanders, B. v., 2015. *A sparse reconstruction method for the estimation of multi-resolution emission fields via atmospheric inversion*. Geoscientific Model Development, 8, 1259–1273. https://doi.org/10.5194/gmd-8-1259-2015.

Ray, J., Yadav, V., Michalak, A. M., van Bloemen Waanders, B., and McKenna, S. A.: A multiresolution spatial parameterization for the estimation of fossil-fuel carbon dioxide emissions via atmospheric inversions, Geosci. Model Dev., 7, 1901–1918, https://doi.org/10.5194/gmd-7-1901-2014, 2014.

Hase, N., Miller, S. M., Maaß, P., Notholt, J., Palm, M., and Warneke, T.: Atmospheric inverse modeling via sparse reconstruction, Geosci. Model Dev., 10, 3695–3713, https://doi.org/10.5194/gmd-10-3695-2017, 2017.

Jones, T.S., Franklin, J.E., Chen, J., Dietrich, F., Hajny, K.D., Paetzold, J.C., Wenzel, A., Gately, C., Gottlieb, E., Parker, H. and Dubey, M., 2021. Assessing urban methane emissions using column-observing portable Fourier transform infrared (FTIR) spectrometers and a novel Bayesian inversion framework. *Atmospheric Chemistry and Physics*, *21*(17), pp.13131-13147.

Lauvaux, T., Miles, N. L., Deng, A., Richardson, S. J., Cambaliza, M. O., Davis, K. J., Gaudet, B., Gurney, K. R., Huang, J., O'Keefe, D., Song, Y., Karion, A., Oda, T., Patarasuk, R., Razlivanov, I., Sarmiento, D., Shepson, P., Sweeney, C., Turnbull, J., & Wu, K. (2016). High-resolution atmospheric inversion of urban $CO_2$ emissions during the dormant season of the Indianapolis Flux Experiment (INFLUX). *Journal of geophysical research. Atmospheres : JGR*, *121*(10), 5213–5236. https://doi.org/10.1002/2015JD024473.

Mallat, S., 1999. *A wavelet tour of signal processing*. Elsevier.

---

## Author Response (AR2)

Dear Editor and Reviewers,

We are very grateful for your comments and suggestions, which have helped to improve our manuscript significantly. We have revised the manuscript accordingly, and the changes can be found in the track-changes file. The following is a point to point response to your comments and suggestions. Corresponding changes in the manuscript are also made available below at the appropriate places, if applicable.

Sincerely,

Benjamin Zanger and Jia Chen on behalf of all co-authors.
* * *
**Editor and Reviewer #2:**

**On line 298, in the sentence "Even though, for both domains ...", the comma is not needed.**

Thank you for the remark, we changed that in the text.

**Throughout the text, the word "relative" has been abbreviated as a rel. I think it should be spelled out (the typesetter might do that anyway before the paper appears in print).**

We changed that throughout the text, and in the axes labels of the figures. The changes in the figures are not found in the track changes.

**In the abstract, the authors state, "A common top-down approach for solving this problem is Bayesian inversion that uses a given Gaussian distributed prior emission field. However, such an approach has drawbacks when the assumed spatial emission distribution is incorrect." I worry that this statement could be misinterpreted by readers. Technically speaking, traditional Bayesian inverse models can be used to correct the spatial (and/or temporal) distribution of emissions. Rather, this approach typically assumes that the correction follows a multivariate normal distribution, and this assumption may not be appropriate for all inverse problems, like the urban applications discussed in this paper.**

Thank you for this remark. We adapted the sentence in the following way:

| Lines 3 ff: | *A common top-down approach for solving this problem is Bayesian inversion  with the assumption of a  multivariate Gaussian  distribution for the prior emission field. However,* |
|---|---|

| | *such an  assumption has drawbacks when the assumed spatial emissions  are incorrect or not Gaussian distributed.* |
|---|---|

**Line 26: Numerous inverse modeling studies also enforce temporal correlation.**

Thank you for this remark. We added this to the text.

| Lines 26 ff: | *Instead, sectors (Jones et al., 2021), spatial correlations (Wesloh et al., 2020), and/or temporal correlations (Jones et al., 2021; Wesloh et al., 2020) are used to construct alternative parameterizations of emission fields to prevent overfitting.* |
|---|---|

**Line 41: The description of Hase et al. (2017) could be confusing to some readers. E.g., it's not clear what "limitations" are being referred to or what "multiple parameterizations" refers to. Hase et al. (2017) use a dictionary that includes different types of spatial patterns or basis functions. As a result, they can represent a diversity of spatial patterns in the emissions field. (I'm not sure if my description is any less confusing, but I think it's another way to describe the important features of their work.)**

Thank you for this remark.
We modified the text to make it clear that this approach increases the sparsity of the emission field and the representation of the field is not unique:

| Lines 26 ff: | *Hase et al. (2017) demonstrated sparse reconstruction with enforced positive emissions estimates of anthropogenic $CH_4$ emissions from synthetic observations in the US.  To increase the sparsity of the emission field, Hase et al. (2017) used a redundant dictionary representation, where  the representation of the emission field is not unique.* |
|---|---|

**Lines 86-89: This description of maximum likelihood could be confusing to some readers. E.g., "From this posteriori distribution a parameter estimation can be made using a maximum likelihood (ML) detector on the a posteriori." It's not clear what parameters this sentence is referring to. The word "detector" here also feels a bit confusing to me (i.e., it implies that ML is being used to detect something). Here's how I think of this process: inverse modelers estimate emissions by maximizing a likelihood function or equivalently, minimizing the negative log of this function. The**

**resulting estimate is often referred to as the ML estimate of the emissions. This likelihood function is based on Bayes Theorem (which leads into Eq. 2).**

Thank you for this remark. We have the same understanding of this process, where the likelihood function is the probability (or probability density) of the parameters derived by Bayes' theorem from the prior probability distribution and the measurement probability distribution.
In engineering this is sometimes referred to as maximum a posteriori (MAP) detector, or maximum likelihood (ML) detector on the posteriori.
But a more common expression in statistics is maximum likelihood (ML) estimation on the posteriori or maximum a posteriori (MAP) estimation.
As parameters, we understand the coefficients describing the emission field.
We adapt our text to be compliant with the literature of statistics and inverse modeling:

| Lines 26 ff: | *From this posteriori distribution an estimation of the parameters, and therefore the emission field,  can be made using a maximum likelihood (ML)  estimator on the a posteriori. Since the ML  estimator acts on the a posteriori, this is commonly referred to as Maximum a posteriori (MAP)  estimator.* |
|---|---|